# Combined Transcriptome Analysis Reveals the Ovule Abortion Regulatory Mechanisms in the Female Sterile Line of *Pinus tabuliformis* Carr.

**DOI:** 10.3390/ijms22063138

**Published:** 2021-03-19

**Authors:** Zaixin Gong, Rui Han, Li Xu, Hailin Hu, Min Zhang, Qianquan Yang, Ming Zeng, Yuanyuan Zhao, Caixia Zheng

**Affiliations:** 1College of Biological Sciences and Technology, Beijing Forestry University, Beijing 100083, China; gongzaixin@bjfu.edu.cn (Z.G.); hanrui7788@163.com (R.H.); xuli1@bjfu.edu.cn (L.X.); huhailin@bjfu.edu.cn (H.H.); zhangmin1990@bjfu.edu.cn (M.Z.); yangqianquan@bjfu.edu.cn (Q.Y.); zengming1990@bjfu.edu.cn (M.Z.); 2Beijing Advanced Innovation Center for Tree Breeding by Molecular Design, Beijing Forestry University, Beijing 100083, China; 3Guangdong Academy of Forestry, Guangzhou 510520, China

**Keywords:** *Pinus tabuliformis* Carr., ovule abortion, single-molecule real-time sequencing, alternative splicing, auxin response, energy metabolism

## Abstract

Ovule abortion is a common phenomenon in plants that has an impact on seed production. Previous studies of ovule and female gametophyte (FG) development have mainly focused on angiosperms, especially in *Arabidopsis thaliana*. However, because it is difficult to acquire information about ovule development in gymnosperms, this remains unclear. Here, we investigated the transcriptomic data of natural ovule abortion mutants (female sterile line, STE) and the wild type (female fertile line, FER) of *Pinus tabuliformis* Carr. to evaluate the mechanism of ovule abortion during the process of free nuclear mitosis (FNM). Using single-molecule real-time (SMRT) sequencing and next-generation sequencing (NGS), 18 cDNA libraries via Illumina and two normalized libraries via PacBio, with a total of almost 400,000 reads, were obtained. Our analysis showed that the numbers of isoforms and alternative splicing (AS) patterns were significantly variable between FER and STE. The functional annotation results demonstrate that genes involved in the auxin response, energy metabolism, signal transduction, cell division, and stress response were differentially expressed in different lines. In particular, *AUX/IAA*, *ARF2*, *SUS*, and *CYCB* had significantly lower expression in STE, showing that auxin might be insufficient in STE, thus hindering nuclear division and influencing metabolism. Apoptosis in STE might also have affected the expression levels of these genes. To confirm the transcriptomic analysis results, nine pairs were confirmed by quantitative real-time PCR. Taken together, these results provide new insights into ovule abortion in gymnosperms and further reveal the regulatory mechanisms of ovule development.

## 1. Introduction

Sexual reproduction is the most evolved reproductive mode in plants, as well as a step in the life cycle of seed plants. The completion of sexual reproduction depends on the normal development of reproductive organs and embryos. Female sterility is a common phenomenon during sexual reproduction in the plant kingdom [1,2], which might result in yield reduction, affect fine variety breeding, and decrease the economic benefit of agronomically important crops and woody plants. Female sterility may be induced by many factors, such as abnormal ovule development, the wrong number of polar nuclei, meiosis, and mitotic disorder. For example, female sterile mutants of *Punica granatum* L. were mainly caused by abnormal ovule development [3]. Awasthi et al. reported that unusual mitotic divisions with a reduced number of polar nuclei in megagametophytes led to female sterility in *Oryza sativa* L. [4]. Furthermore, Rosellini et al. found that incomplete meiosis resulted in a lack of embryo sack development and the accumulation of callose in the nucellus in female sterility lines of *Medicago sativa* L. [5]. In addition, abnormal female gametophyte formation was the main reason for ovule abortion, which was observed in female sterility lines of *Allium sativum* L. [6]. These studies suggested that the mechanism of female sterility was diverse; thus, a better understanding of the molecular regulation of female sterility is essential for improving germplasm resources and breeding the plus tree.

In the past few years, many studies about the molecular mechanism of ovule development have been reported, revealing many factors related to female sterility in plants. For example, Pagnussat et al. found that the distribution of auxin during female gametophyte development regulated the differentiation and conversion of gametic and other cells, affecting the embryo sac development in the initial stage [7]. Reactive oxygen also played a role in embryo sac patterning and fertilization, which influenced the development of female gametophytes by its concentration [8]. Moreover, ribosomal protein mutations might lead to ribosomal insufficiency, and the expression level of ribosomal protein genes was related to the number of seeds and defective ovules [9]. Studies on female plant sterility mainly concentrated on herbs with a short life cycle, and less on woody plants, especially gymnosperms.

Next-generation sequencing (NGS) is able to quantitatively analyze transcripts correctly, but the short-read length limits the accuracy of gene structure prediction. However, single-molecule real-time (SMRT) sequencing such as the Pacific Biosciences platform (PacBio) can produce longer reads without assembly. Depending on the long reads of SMRT, data sets can be excellent at connecting different exons, up to entire transcripts [10,11]. This method provides preferable conditions for the analysis of alternative splicing (AS) and simple sequence repeats (SSRs) [12,13]. Increasing evidence suggests that AS can change the structure of transcripts and proteins. When AS is activated in the initial regulatory period, it can effectively control the entire developmental pathway and quantitatively regulate gene expression [14,15]. AS is involved in most plant development processes and the response to environmental stress [16]. For example, several studies have suggested that AS is related to the vegetative growth of plants in many species, including *Arabidopsis thaliana* [17], *Salvia miltiorrhiza* [18], and *Astragalus membranaceus* [19]. Furthermore, Sun et al. indicated that the fifth intron being retained caused the deletion of the sixth exon of ohms1 (open hull and male sterile 1), which led to male sterility in rice [20]. In addition, SSRs are tandem repeat nucleotides in DNA sequences, which are often used as genetic markers and play important roles in the regulation of gene expression and the development of organisms [21]. Dou et al. determined the location of the gene that led to female sterility in wheat by SSRs [22]. Kato used SSRs to identify the position of *ST8*, which influences male and female sterility in soybeans [23]. However, there is no such relevant research in gymnosperms.

Gymnosperms retain some characteristics of spore plants and have the evolutionary characteristics of angiosperms, which have the most complex sporophytes and the most reduced gametophytes [24]. Therefore, as the node of plant evolution, gymnosperms have essential research value, especially in terms of reproductive development [25]. *Pinus tabuliformis* Carr., the main afforestation species in north China, is an endemic and evergreen tree species [26]. It is noteworthy that *P. tabuliformis* is an excellent material for the study of the molecular regulation of gymnosperm ovule development, because its ovule takes about 13 months to complete cellularization [27]. Our previous study showed that line No. 28, in a *P. tabuliformis* seed orchard in Xingcheng, Liaoning, China, was a female sterile mutated line. The appearance of the cone was normal but the free nuclear mitosis of megagametophyte (FNMM) was terminated at an early stage, which caused ovule abortion and led to the abnormal production of seeds [28]. To reveal the mechanism of ovule abortion, our research team performed an analysis based on the platform of the NGS technologies and screened a series of genes relevant to mitosis and phytohormones [29]. However, in order to further reveal the mechanism of ovule development, sufficiently long transcripts are required, which is the most significant shortcoming of NGS.

In the present study, to extend our knowledge of the mechanism of *P. tabuliformis* female sterility during the FNM process, we combined NGS and SMRT sequencing to generate a more complete full-length transcriptome of *P. tabuliformis*, which was conducive to the qualitative and quantitative analysis of isoforms and more accurate analysis of AS events. Firstly, three different development stages of fertile line ovules and sterile line ovules of *P. tabuliformis* were measured and compared. Combined sequencing was employed to identify transcripts and to separately examine their expression patterns in different lines, screening out potential differentially expressed genes (DEGs) participating in the regulation of ovule abortion. These DEGs were enriched in energy metabolism, signal transduction, cell division, and stress response pathways, and the expression level was validated by quantitative real-time PCR. In addition, the AS and SSRs of the mutant and normal ovules in *P. tabuliformis* were explored according to SMRT sequencing. Based on these results, the molecular regulation mechanism of the FNMM half-stopping and ovule abortion in a female sterile mutate line in *P. tabuliformis* was further revealed, and our comprehensive analyses afford new insight into the molecular mechanisms of ovule development in gymnosperms.

## 2. Results

### 2.1. Cone and Ovule Phenotype Analysis of P. tabuliformis

The size of the cone and scale in two lines was almost the same in the early stage of FNMM, up to the middle stage, while the cone and scale of the female sterile line (STE) were slightly larger than in the female fertile line (FER) (Figure 1A–C). Until the late stage, the cone and scale of STE were significantly developed, and obviously larger than in FER. However, the size of the ovules was similar between FER and STE in each development stage (Figure 1A–C). However, the FER and STE ovules were of similar volume from FNM1 to FNM3 (Figure 1D). The morphological analysis of ovules at each of the sampling stages is described in Figure 1E. In the early stage of FNMM, the anatomical structure of STE and FER ovules was similar, and the megagametophyte consisted of a large vacuole and dozens of free nuclei (Figure 1E). In the middle stage, the free nuclei were still increased in FER; however, this trend was arrested in STE (Figure 1E). Up to the late stage, the number of free nuclei was sharply increased and filled the center in FER ovules, while the free nuclei in STE ovules began to disappear and their centers were empty (Figure 1E). The differences of the cone and ovule phenotypes of *P. tabuliformis* between FER and STE occurred from the FMN2 stage, and the development of FG was blocked in STE.

### 2.2. Combined Sequencing Approach to Analyze the Ovule of P. Tabuliformis

To investigate the process of FNM in ovule development in female fertile and sterile lines of *P. tabuliformis*, and reveal the reasons for FNMM discontinuation, we selected three different time points, including early February (the early period of the FNM process), early March (the middle period of the FNM process), and early April (the late period of the FNM process). Female fertile line (FER) ovule samples at the collection periods described above were marked as FER-FNM1, FER-FNM2, and FER-FNM3, respectively, while female sterile line (STE) ovule samples were marked as STE-FNM1, STE-FNM2, and STE-FNM3, respectively, and each sample from the above periods was in triplicate. Eighteen mRNA samples from different stages of ovules were subjected to 2 × 150 paired-end sequencing using the HiSeq^TM^ 4000 platform, with 961,064,004 reads produced (Figure 2). The repeatability of the RNA-seq libraries was evaluated using principal component analysis, and the reproducibility of the libraries was satisfactory (Appendix A). Then, they were combined with equal amounts of total RNA from the three stages (FNM1, FNM2, and FNM3). Finally, we obtained two pooled libraries (FER and STE). Two libraries were normalized and subjected to SMRT sequencing using the PacBio RS II platform (Appendix A). In total, 276,887 (FER) and 307,741 (STE) ROIs were generated (Figure 2), and most of them (73.27% in FER, 64.33% in STE) were full-length nonchimeric (FLNC) reads, which had the entire transcript region from the 5′ primer to the 3′ primer, and a 3′ poly(A) tail was observed (Figure 3A). In STE, there were more FLNC less than 2000 bp, while in FER, there were more FLNC longer than 2000 bp (Appendix A).

As shown in Appendix A, there was a significant variation in transcript length distribution between the two platforms, and the read length was longer for PacBio; only 1.5% of the transcripts from the PacBio reads were <600 bases, but nearly 60% of the assembled transcripts from the NGS reads were in this range. Based on the Illumina short-read data, the length of N50 was 1767 bp, whereas N50 was 3564 bp on the PacBio platform, demonstrating that the data assembly by PacBio was better than that by Illumina. Through SMRT sequencing, 19.07 GB of raw data were produced from the two libraries (FER and STE), and 81,524 unique full-length transcripts were obtained through pipeline analysis. Moreover, 0.96 billion pairs of clean reads were produced from 18 libraries using NGS to calculate the expression level (reads per kilobase of exon model per million mapped reads, RPKM) of each full-length transcript, with a mapping ratio of 85.43–87.99%. Therefore, NGS reads corrected using SMRT subreads were more valuable than relying on a single platform’s data. Combined sequencing is also more worthwhile for species without a reference genome.

### 2.3. Functional Annotation of Full-Length Transcriptomes in P. Tabuliformis

For the annotation of genes, alignment searches were conducted against public bioinformatics databases. To further evaluate the function distribution of our transcriptome, we used Gene Ontology (GO) annotation to classify the full-length transcripts obtained by SMRT in FER and STE (Appendix A). Annotated genes were divided into three categories: biological process, cellular component, and molecular function. In the biological process category, both in FER and STE, annotated genes were significantly enriched in metabolic processes, cellular processes, and single-organism processes. Furthermore, cells, cell parts, and organelles were more abundant in the cellular component in each line; in terms of molecular function, genes from different lines were enriched into similar processes, mainly for catalytic activity and binding.

The Kyoto Encyclopedia of Genes and Genomes (KEGG) was also used to investigate the pathway and network of the molecular regulation of FER and STE. In both FER and STE, annotated genes were mapped into 129 pathways (Appendix A). As with GO annotations, the KEGG pathway classes between the two lines of terms were very similar. In both FER and STE, metabolic pathways, biosynthesis of secondary metabolites, biosynthesis of antibiotics, and microbial metabolism in diverse environments were the top four pathways with the most abundant transcripts, but compared with FER these four pathways enriched more genes in STE (Appendix A). The number of genes enriched in FER and STE were slightly different in each pathway, but there were no significant differences in terms of pathway annotation. Based on the functional annotation and classification results of GO and KEGG, we obtained a thorough functional characterization for the full-length transcripts of the FER and STE ovule. The results of GO and KEGG also showed that the functional annotation and classification of transcripts between the two lines were generally similar, indicating that the pathway level was conservative for the transcriptome of FER and STE in *P. tabuliformis*.

### 2.4. Candidate Genes with Opposite Expression Patterns during the FNM Process in STE and FER

To find the DEGs that might have influenced the development of FG, we counted the DEGs of each FNM period and these expression levels in FER and STE. From Figure 3B, we found in all stages that there were more DEGs upregulated in FER than downregulated. There were about 25,000 DEGs in each FNM period. To obtain the most significant DEGs, according to the variety of gene expression during the whole development process in FER and STE, genes generated from NGS and corrected by SMRT could be clustered into eight profiles by Short Time-series Expression Miner (STEM) software. These model profiles were chosen to summarize the expression pattern of the genes. Among the eight patterns, we identified four patterns of genes that showed very significant *p*-values (colored boxes) (Appendix A); these four profiles were significant in both FER and STE, containing two downregulated patterns (profiles 1 and 0) and two upregulated patterns (profiles 6 and 7). Genes in profiles 1 and 0 in STE were combined; similarly, profiles 6 and 7 were also combined in FER. Twenty-three DEGs that were upregulated in FER but downregulated in STE (part 1) were screened out (Figure 3C). Sixty-nine DEGs that were upregulated in STE but downregulated in FER (part 2) could also be filtered out (Figure 3D). The DEGs expressed in these two parts showed contrary trends in FER and STE during the FNM process, which could probably play a pivotal role in the development of FG.

Two segments of genes were subjected to GO term analysis. They were still classified into three main categories, including “cellular component,” “molecular function,” and “biological process” (Figure 3E,F). In the biological process category, metabolic process, cellular process, and single-organism process were the most significant in two parts. In the cellular component category, cell and cell parts were more obvious. Most DEGs were in catalytic activity and binding two parts in the molecular function category. Then, DEGs were subjected to KEGG pathway enrichment analysis. The KEGG pathways were mapped and are shown in Appendix A. In part 1, annotated genes were enriched in 14 KEGG pathways and 24 pathways in part 2.

In combination with the two parts of DEGs, Figure 4 and Appendix A list the DEGs with significant differences in expression levels among the different lines, which may be related to the female sterility of *P. tabuliformis*. Energy metabolism and signal transduction are important for FNMM development. Therefore, we investigated the genes associated with carbohydrate metabolism and signal transduction, including sucrose synthase, phosphofructokinase, chorismate synthase, clavata 1-like protein, tetraspanin, reticulons, and plasma membrane ATPase-related genes. Among these DEGs, most showed significantly lower transcript levels in STE ovules. Specifically, the transcript levels of sucrose synthase genes *SUS* (Iso_0064722, Iso_0065236, Iso_0045687), *AGPL1* (Iso_0015412), clavata 1-like protein-related genes *CLV1* (Iso_0025308, Iso_0025527, Iso_0073847, Iso_0079334), and some other transmembrane protein-related genes *RTNLB8* (Iso_0006259), *TET18* (Iso_0006259), *NPF3.1* (Iso_0006259) were markedly lower in STE compared to FER.

There were also several DEGs related to mitosis and apoptosis, encoding cyclin (CYCB, Iso_0016253), tubulin (TUBA, Iso_0012780), dihydrofolate reductase (DHFR, Iso_0053543), and cysteine proteinase inhibitor (CPI, Iso_0031083), which, differentially expressed between FER and STE, also might lead to FNM half-stop. DEGs associated with mitosis like *CYCB*, *TUBA*, and *DHFR* all kept a high expression level in FER and a low one in STE. Similarly, the expression of *bHLH66* (Iso_0067886) was obviously different between FER and STE. Furthermore, the transcript levels of several genes related to the stress response, such as *PER* (Iso_0053653, Iso_0049597, Iso_0010250), *SOBIR1* (Iso_0024626), *exgA* (Iso_0058895), and *CALS3* (Iso_0078524), showed opposite expression levels in varying lines.

Auxin has been implicated in nuclear division and cell growth. Based on the sequencing data, we found that the expression levels of genes related to auxin were different in FER and STE of different development stages. As shown in Figure 5, we identified 10 DEGs encoding auxin signaling components, with four of them encoding auxin-responsive protein (*IAA*, Iso_0007499, Iso_0022099, Iso_0023965, Iso_0024690), four encoding auxin response factor 2 (*ARF2*, Iso_0009878, Iso_0016698, Iso_0018834, Iso_0025193), and two encoding transport inhibitor response 1 (*TIR1*, Iso_0015783, Iso_0060648). The abovementioned genes were all upregulated in FER and downregulated in the STE ovules; IAA27 was 4-fold higher in the FER in the FNM3 stage, and TIR1 was also significantly highly expressed in the FNM3 stage.

### 2.5. AS Analysis of DEGs with Different Expression Patterns during the FNM Process in Two Lines

Describing the complexity of AS at the transcriptome scale is one advantage of full-length sequencing. To analyze the AS events of transcript isoforms, full-length transcripts were partitioned into gene families based on their k-mer similarity, then reconstructed into a coding reference genome based on k-mer similarity (Figure 6A). A total of 42.69% had one isoform in FER and a slightly lower percentage (36.91%) in STE, and there were more contigs with more than 15 isoforms in STE (1.28%, 186) compared with FER (1.33%, 180) (Figure 6C). Seven AS modes (skipping exon, mutually exclusive exons, alternative 5′ splice-site, alternative 3′ splice-site, retained intron, alternative first exon, and alternative last exon) existed in the ovule of *P. tabuliformis* (Figure 6B). In total, 1830 and 1243 AS events were identified in FER and STE, respectively. In each AS mode, there were more AS events in FER, and the retained intron (RI) was the most significant AS mode both in FER (51.09%) and STE (48.91%), which was also the predominant mode in plants [30]. Together with alternative 5′ and 3′ splice-site AS events, these three types of AS event accounted for over 80% of detected events. On the contrary, mutually exclusive exons was the least frequent AS event (Figure 6D).

To explore whether the existence of AS for the genes might affect female sterility in *P. tabuliformis*, we researched AS in the DEGs, analyzed as above with the opposite expression patterns during the FNM process in varying lines. Finally, isoforms of *AGPL1* (associated with starch metabolism), *bHLH66* (associated with plant growth and development), and *TUBA* (associated with mitosis) were diverse in FER and STE (Appendix A); RI was the main AS event and existed in all three DEGs, while *TUBA* also had A3 splicing (Figure 6E, Appendix A). The number of isoforms varied between FER and STE in *AGPL1* and *TUBA*, but *bHLH* had only one isoform in each line; however, the splicing isoforms were different.

### 2.6. Development and Characterization of SSR Markers

To investigate the differences in regulatory mechanisms between FER and STE ovules, we discovered the characteristics of SSR markers in two lines. Although we can develop SSR markers based on NGS, NGS generally requires the assembly of short RNA-seq reads, so transcriptomic sequences constructed using NGS may be misassembled. Using SMRT sequencing, intact RNA molecules can be sequenced without the need for fragmentation or post-sequencing assembly; thus, we can obtain information for SSRs more accurately [31]. In this study, 40,828 (in FER) and 44,195 (in STE) genes were generated to discover potential microsatellites and were defined as di- to hexanucleotide motifs (Table 1). Furthermore, the 5530 potential SSRs in FER and 5096 in STE were further analyzed. As shown in Table 1, the dinucleotide repeats (45.5% in FER, 49% in STE) ranked first; next came the tri- (38.4% in FER, 36.4% in STE), tetra- (8.4% in FER, 8% in STE), hexa- (5.2% in FER, 4.5% in STE) and pentanucleotide (2.4% in FER, 2.6% in STE) repeats. Moreover, the number of repeat units from Dinucleotide to hexanucleotide motifs were summarized. As shown in Appendix A and Figure 7A,B, the most represented repeat units of potential SSRs was 4–7, which accounts for 74.8% in FER and 72.2% in STE. Trinucleotide repeats were the most abundant in this part, in both FER and STE. FER had more potential SSRs than STE, and the difference in quantity of 4–7 repeat units between FER and STE was more significant than the other levels of the repeat unit. Based on Figure 7C,D, we found that the SSR motifs in the two lines shared significant similarity. The AT/TA dinucleotide repeat was the most abundant motif in FER (30.4%) and STE (32.1%), followed by AG/CT (11% FER, 12.1% STE), AGC/CTG (8.9% FER, 6.7% STE), AAG/CTT (7.5% FER, 7% STE), AAT/ATT (6% FER, 6.7% STE), and ATC/ATG (5.9% FER, 5.6% STE). The six types of nucleotide repeats mentioned above represented about 70%.

### 2.7. Verification of the Gene Expression Profile by qRT-PCR

To confirm the transcriptomic analysis results, DEGs were randomly selected for quantitative real-time PCR (qRT-PCR) validation using the same type of samples compared with formerly used samples in NGS analysis, including *SUS*, *PER12*, *PFK2*, *CLV1*, *SOBIR*, *UBQ10*, *TUBA*, *AGPL1*, and *TET18*. The primers of these DEGs are listed in Appendix A. The expression profiles of the candidate DEGs revealed by the qRT-PCR data were consistent with those derived from sequencing. As shown in Figure 8, the energy metabolism-related genes *SUS*, *AGPL1*, and *PFK2* were highly expressed in FER, the expression level of *PKF2* was significantly high in the FNM3 stage, and *SUS* and *AGPL1* were high in FNM1. In addition, the expression of *CLV1* and *TET18*, which are associated with signal transduction, and *TUBA* were 61.4-fold lower in STE ovules compared with FER ovules. Additionally, the expression levels of *PER12* and *UBQ10* related to the stress response were higher in FER than STE, especially in the FNM1 stage. On the contrary, *SOBIR1* was obviously upregulated in STE, which is associated with apoptosis, especially in the FNM2 stage, and the expression level of *SOBIR1* was 10.2-fold higher in STE than FER. The expression profiles of the candidate DEGs revealed by qRT-PCR data were consistent with those derived from sequencing, and further confirmed the differences in the energy metabolism, signal transduction, stress response, and apoptosis between FER and STE.

## 3. Discussion

The development of the FG is one of the key processes of the alternation of generations for gametophytes and sporophytes in plants, which decides the future fate of the seeds. To date, plenty of studies of FG development during FNM have been based on angiosperms, especially in *Arabidopsis* [32]. Few reports have focused on female sterility in woody plants, such as *Ulmus minor* Mill. [33] and *Xanthoceras sorbifolium* Bunge [2]. However, due to the ovule mutants of gymnosperms being rare and the ovules being small and difficult to obtain, the molecular regulation of FG development remains largely unknown in gymnosperms. Ling et al. reported that the siliques of FG defective lines were significantly smaller than those of the wild type in *Arabidopsis* [34]. In the present study, the size of the cone and scale were similar in two lines in the FNM1 stage. However, up to the FNM2 stage, STE were slightly larger than those of the FER, but significantly larger than those of the FER at the FNM3 period, indicating that the growth of cones and scales was more rapid in the STE than in the FER (Figure 1). With the development of the ovule, the mitosis of FG was interrupted in the STE from the FNM2 stage. As a result, the viability of STEs might be reduced, and their growth and development was blocked.

Phenotypic changes were usually caused by changes at the molecular level; thus, an accurate panorama of gene expression at the transcriptional level is particularly important. In this investigation, we combined second-generation sequencing (Illumina) and third-generation sequencing (PacBio) to analyze the regulatory mechanisms of ovule abortion in *P. tabuliformis* (Figure 2). Previous studies about female sterility mainly used NGS [35,36]; however, NGS was limited for reads of short length as the sequencing products were incomplete and required assembly. SMRT sequencing can overcome the disadvantage of NGS because of its long reads [14]. NGS reads were corrected by SMRT subreads, which helps to more accurately analyze the transcriptome differences between FER and STE, qualitatively and quantitatively. We found that, no matter the average length or the N50 length, it was obvious that the full-length transcripts were much longer than the de novo assembled transcripts (Appendix A). Therefore, the transcripts had different periods in different lines of *P. tabuliformis*, and combined sequencing was more worthwhile than relying on a single platform.

To obtain the functional distribution of our transcriptome, GO annotation and KEGG pathway analysis were used to annotate the genes in FER and STE. The results of GO annotation and the classes of the KEGG pathway were quite similar in varying lines (Appendix A), indicating that the transcriptome of FER and STE in *P. tabuliformis* was conservative at the pathway level. To further reveal the factors related to female sterility, we analyzed the expression of genes. In this study, we obtained more than 20,000 DEGs in each FNM stage. To obtain the most significant DEGs, we selected them from Figure 3D,E, which has opposite expression patterns between FER and STE during the FNM process. During the FNM process, a total of 99 DEGs were upregulated in one line and downregulated in the other, which might be the core factor of ovule abortion in *P. tabuliformis*.

A previous study revealed that AS participates in the reproductive development of *Arabidopsis* [37]. AS has been reported to regulate the formation of male and female gametogenesis in *Arabidopsis* [38], which also led to male sterility in rice [20]. In this study, we detected seven AS modes in the ovule of *P. tabuliformis*; there were more AS events in the FER than in the STE ovule (Figure 6B,D). We further analyzed the DEGs with opposite expression patterns during the FNM process between FER and STE and found that the isoforms of *AGPL1*, *bHLH66*, and *TUBA* were different in the two lines (Figure 6E). We speculated that the quantitative differences in AS events, especially the different isoforms of *AGPL1*, *bHLH66*, and *TUBA* between the two lines might lead to the development of abnormal ovules in STE. AS will be an important research topic in future studies. In addition, developing SSR markers is another merit of SMRT sequencing. SSRs have been used for determining protein function, genetic development, and gene expression regulation [21]. Several studies on gymnosperms have already focused on SSRs, such as *Taxaceae*, *Pinus massoniana*, and *Ginkgo biloba* L. [39,40,41]. The potential SSRs identified in this research provided productive resources for further investigation of SSRs in *P. tabuliformis* and established a foundation for the determination of molecular markers in gymnosperms, which could be valuable in molecular studies in this unique species in the future.

Auxin is a mobile signaling molecule regulating various processes in plant development; it is inevitable that a gradient forms during polar auxin transport [42]. Efflux and influx carriers regulate the distribution of auxin and influence the development of ovules [43]. Robert et al. also found that a local auxin source coupled to feedback regulation of auxin transporter facilitators’ polarity in the embryo is sufficient to generate a robust auxin gradient that instructs the formation of the embryo in *Arabidopsis* [44]. Furthermore, it has been reported that the reduction of auxin biosynthesis might lead to ovule loss in *Arabidopsis* [7]. *AUX/IAA* was the auxin primary response gene, suggesting that the correct expression of auxin response genes is necessary for FG development [45]. *ARF* binds to the promoter region of the auxin-responsive element to stimulate or repress transcription, and the downregulation of *ARF6* and *ARF8* will induce the ensuing arrest of FG development [46]. Furthermore, previous studies reported that *TIR1* is an auxin receptor that mediates *AUX/IAA* degradation and auxin-regulated transcription [47]. In this study, *AUX/IAA*, *ARF2*, and *TIR1* were all significantly upregulated in FER and downregulated in STE, indicating that auxin might be insufficient in the STE ovules, thus hindering the nuclear division and influencing the metabolism in FG. The low expression of *TIR1* showed that the auxin receptor was limited in the STE ovules; the residual auxin was transported from ovule to scale, and the accumulation of auxin in scale might be a main reason for the size difference between FER and STE.

Energy metabolism regulates vital movement at the whole plant level, and so is a critical pathway during reproductive growth. Several regulatory genes of starch and sucrose metabolism and glycolysis have been identified as associated with reproductive development. *SUS* regulates the number of bud primordia and accelerates bud growth [48]. Kong et al. found that the expression level of *SUS* and *AGPL* influences the number of seeds, and these are significantly low in male fertile lines of rice [49]. *PFK2* encodes 6-phosphofructokinase, which is the most important element for the regulation of glycolysis [50,51]. In the present study, many genes encoding components associated with starch and sucrose metabolism and glycolysis showed different expression patterns during the FNM process in FER compared with STE. The expression levels of *SUS* and *AGPL1* were higher in the FER ovules than in STE during the whole development, and especially highly expressed in the FNM1 stage; the expression level gradually reduced as the FG developed. *PFK2* was upregulated from FNM1 to FNM3, and significantly less expressed in STE than in FER. This indicated that the development of FG is a process of energy dissipation; respiration expends starch and sucrose to provide energy for free nuclear mitosis. With the development of FG, glycolysis was gradually enhanced, and thus the starch and sucrose were consumed.

In recent years, many genes related to inter- and intracellular transport that leads to ovule abortion have been reported. Huck et al. found that the signaling process was pivotal to ensure the ovule develops normally in *Arabidopsis*, and signal transduction was damaged in the female semi-sterile mutant [52]. *TET* genes control the cell proliferation, cell differentiation, and cell identity, moreover, and are also expressed in reproductive tissues and affect FG development in *Arabidopsis* [53,54]. RTNLB contributes significantly to endoplasmic reticulum (ER) modeling, which plays an important role in terms of linking the cortical ER to the plasma membrane and intracellular trafficking [55,56]. In this study, the expression levels of *TET18* and *RTNLB8* were markedly higher in FER than in STE, of which *TET18* was upregulated during the ovule development, showing that with the mitosis of FG, the intercellular communication was more vigorous. *RTNLB8* was significantly highly expressed in the FNM1 stage in FER ovules; therefore, we predicted that, in the early stage, ER would be produced in a large volume, which was the preparation of the nuclear membranes for mitosis. Kayes and Clark found that in *Arabidopsis*, *CLV* genes can control vegetative and reproductive growth [57], and regulate the number of carpels and seeds [58]. Our results showed that *CLV1* was highly expressed in FER, especially during the FNM1 period, and less expressed during the whole development period in the STE ovule. We predicted that the low expression of *CLV1* could result in cell signaling impairment and a failure of division and differentiation, which impeded FG development.

Cell division is a key step in the growth and development of an organism, especially in terms of reproductive development. Previous studies found that several regulatory genes of mitosis have been identified. For example, *CYCB1;1* was highly expressed and reached the peak in the M phase and degraded upon M phase exit, thus controlling entrance and retreat from the M phase in *Nicotiana sylvestris* [59]. *TUBA* participates in mitosis, chromosome movement, and nuclear movement, and mutations destabilize spindle microtubules in *Aspergillus nidulans* [60]. Several studies reported that *DHFR* was highly expressed at the G1/S of the cell cycle, which increased as cells underwent DNA synthesis and decreased as cells entered quiescence or terminally differentiated [61,62]. In our investigation, with FNMM developing, *CYCB* and *TUBA* were significantly upregulated in FER, and *DHFR* was also highly expressed in the FNM1 period. On the contrary, they were all downregulated in STE. It is suggested that mitosis may be arrested at the G2/M phase, and DNA and some other biomolecules’ insufficient accumulation in G1/S, together with spindle formation, was abnormal, which led to free nuclear division being difficult in STE.

Apoptosis is autonomic cell death to maintain normal physiological function, and is regulated by a series of related molecules in the cell. Gao et al. found that the overexpression of *SOBIR1* leads to the activation of cell death [63], while a mutation can repair the defects in the Golgi structure and prevent the premature shedding of floral organs, as well as participate in H_2_O_2_ regulation in *Arabidopsis* [54,64]. Our results showed that *SOBIR1* was upregulated in STE, especially in the FNM3 stage, which might induce FG apoptosis and impact vesicle transportation, and also affect the accumulation of H_2_O_2_. However, the high expression level of *SOBIR1* in STE might also be the result of ovule abortion; the apoptosis that occurred in the FNM1 period led to the vicious circle by which the development of FG was damaged. We found that many genes that regulated the essential pathways of FG developing were downregulated in STE, which also may be induced by apoptosis. The accumulation of H_2_O_2_ led to a marked increase in lignin content and accelerated cell death [65,66]. *PER* regulates a variety of oxidation–reduction reactions to scavenge reactive oxygen, avoiding oxidative stress and participating in resistance to environmental stresses [67]. To resist the abiotic stresses, callose formed to defend the organism and accelerate the formation of the cell wall [68]. Rosellini et al. found that callose deposition also existed in a sterile female line of *Medicago sativa* L [5]. *CALS3* is responsible for the synthesis of callose in the temporary callose wall of microspores and is essential for exine formation during microsporogenesis in *Arabidopsis* [69]. In this study, we found that *PER* was highly expressed in FER and less expressed in the STE ovule, while *CALS* was highly expressed in STE, and less in FER. Due to the opposite expression levels, we proposed that STE ovules might, under oxidative stress, due to the lack of coordination of active oxygen scavenging and H_2_O_2_ overaccumulation, experience a marked increase in lignin content and generate a series of nonfunctional proteins. Furthermore, in *Arabidopsis*, *CPI* encodes enzymes that have emerged as key in the regulation of programmed cell death (PCD) and help cells to avoid active oxygen damage by inhibiting H_2_O_2_ amplification [70]. We found that *CPI* was nearly absent in STE, which may lead to PCD and the accumulation of active oxygen.

Yang et al. reported that the ovule abortion mechanism of rice was related to starch and sucrose metabolism, plant hormone signal transduction, protein modification and degradation, and oxidative phosphorylation [36]. In *Brassica napus*, DEGs regulating brassinosteroid biosynthesis, adaxial/abaxial axis specification, auxin transport, and signaling were regarded as the reasons for female sterility [52]. Similar to other species, in our study energy metabolism, auxin transport, and signaling were also essential in the ovule abortion mechanism of *P. tabuliformis*. However, during the FNM process, cell division and apoptosis might also be important elements. The reason for this difference may be that free nuclear mitosis is an indispensable step in the ovule formation process in gymnosperms.

## 4. Materials and Methods

### 4.1. Plant Materials and RNA Sample Preparation

Our research group found that the FER ovule and STE ovule were genetically closely related by the DNA marker technique [71]. Therefore, FER and STE ovules were used as suitable materials to study the mechanism of ovule abortion in conifers. Then, we selected three FER trees and STE trees from the *P. tabuliformis* Seed Garden, Xingcheng, Liaoning Province, China. Material collection for this research was permitted by the Forestry Administration of Xingcheng. Cones of similar size were selected from trees growing in common habitats, gathered in early February (the early period of the FNM process), early March (the middle period of the FNM process) and early April (the late period of the FNM process), and samples were collected from the middle of the crown of each tree. The ovules were removed from the scales under a dissecting microscope. Each sample consisted of hundreds of ovules from five cones. These samples were placed into liquid nitrogen overnight, then stored at −80 °C. Paraffin sections were done according to Zhang et al. [72] and we observed the developmental stages of ovules under a microscope (Leica, Weztlar, Germany).

Total RNA was extracted from ground samples using a Plant RNA Assistant Kit (Kebaiao, Beijing, China) according to the manufacturer’s instructions. The integrity of the RNA was quantified with an Agilent 2100 Bioanalyzer (Agilent Technologies, Böblingen, Germany) and agarose gel electrophoresis. The purity and concentration of the RNA were determined with the Nanodrop 2000 (Thermo Scientific, MA, USA).

The total RNA was prepared for two experiments: (1) The RNA-seq samples of FER and STE stages were used to construct 18 libraries (FER-FNM1, FER-FNM2, FER-FNM3, STE-FNM1, STE-FNM2, and STE-FNM3; each sample had three replicates) and were subjected to 2 × 150 paired-end RNA-seq using Illumina HiSeq^TM^ 4000 (Illumina, San Diego, CA, USA). (2) Total RNAs with an RNA integrity number (RIN) value higher than 8 were pooled equally from three development stages (FNM1, FNM2, FNM3) of FER and STE for single-molecule real-time (SMRT) sequencing using the PacBio RS II platform (Pacific Biosciences, Menlo Park, CA, USA).

### 4.2. Analysis of PacBio SMRT

The SMRT Link v5.0.1 pipeline [73], supported by Pacific Biosciences (Menlo Park, CA, USA), was used to classify and cluster the raw sequencing reads of the cDNA libraries into transcript consensus. In short, CCS (circular consensus sequence) reads were extracted and classified into short reads, full-length chimera reads, non-full-length reads, and full-length nonchimeric (FLNC) reads based on cDNA primers and the poly(A) tail signal. Then, the FLNC reads were clustered by Iterative Clustering for Error Correction (ICE) software (Menlo Park, CA, USA) to generate the cluster consensus isoforms. The final transcriptome isoform sequences were filtered with software CD-HIT-v4.6.7 (parameters: -c 0.99, -T 6, -G 0, -aL 0.90, -AL 100, -aS 0.99, -AS 30) to remove redundant sequences using a threshold of 0.99 identities.

### 4.3. Functional Annotation

For annotation, sequences were BLAST analyzed against the NCBI nonredundant protein (Nr) database (http://www.ncbi.nlm.nih.gov), the Swiss-Prot protein database (http://www.expasy.ch/sprot), the Kyoto Encyclopedia of Genes and Genomes (KEGG) database (http://www.genome.jp/kegg), and the COG/KOG database (http://www.ncbi.nlm.nih.gov/COG) with BLASTx program (http://www.ncbi.nlm.nih.gov/BLAST/) at an *e*-value threshold of 1 × 10^−5^ to evaluate the sequence similarity with genes of other species. Gene Ontology (GO) annotation was analyzed by Blast2GO software [74] with Nr annotation results of isoforms, and the functional classification of isoforms was performed using WEGO software [75]. The KEGG database was used to explore the genes involved in the biological pathways. The pathway was defined as a significantly enriched pathway for genes if the *p*-value < 0.05.

### 4.4. Analysis of Differentially Expressed Genes (DEGs)

The reads per kilobase of exon model per million mapped reads (RPKM) was generally used to estimate the gene expression level [76]. The DEGs were identified based on Audic and Claverie’s method [77]. DEGs were identified by IDEG6 software (http://telethon.bio.unipd.it/bioinfo/IDEG6_form/). All statistical tests for this research were calibrated for multiple testing with the Benjamini–Hochberg false discovery rate (FDR). The FDR was less than 0.01, and an absolute value of log_2_ ratio of more than 1 was used to confirm significant differences in gene expression.

The gene expression data for heat map were normalized to Z-score. Z-score normalization was the expression level of each gene minus the average expression level of all genes, then divided by standard deviation.

### 4.5. Alternative Splicing (AS) Detection

To analyze AS events of transcript isoforms, depending on k-mer similarity, we used the COding GENome Reconstruction Tool (Cogent) [22] to divide transcripts into gene families. De Bruijn graph methods were also used to reestablish each family into a coding reference genome. Furthermore, the transcript isoforms analysis of the AS events was by a SUPPA [78] tool.

### 4.6. Simple Sequence Repeat (SSR) Marker Prediction

MISA (http://pgrc.ipk-gatersleben.de/misa/) was used to discover the simple sequence repeats (SSRs) in the whole transcriptome. The minimum number of repeat units is as follows: six for dinucleotides, five for trinucleotides, and four for tetra-, penta-, and hexanucleotides. In addition, the shortest distance between two SSRs is 100 bp, which can be regarded as one SSR.

### 4.7. Real-Time Quantitative PCR Analysis

Total RNA for RNA-seq samples was also used for qRT-PCR in this study. Reverse transcription into cDNA was performed with SuperReal PreMix Plus (TIANGEN Biotech, Beijing, China). All of the experiments were performed following the protocols included with the kits. Primers were designed using Primer Premier 6. The qRT-PCR was performed using a MiniOpticon Two-Color Real-Time PCR Detection System (BIO-RAD, Hercules, CA, USA). The PCR conditions were 95 °C for 30 s, followed by 45 cycles of 95 °C for 3 s for denaturation and 60 °C for 30 s for annealing and extension. The relative expression levels of the target genes were normalized and then calculated using the comparative Ct (2^−∆∆Ct^) method. For each sample, PCR was repeated three times. EF1 (elongation factor 1) was selected as the internal control for normalizing the results, and the FER ovules in FNM1 stage were used as a reference sample whose value was set to 1. The relative gene expression was evaluated using the comparative cycle threshold method.

## 5. Conclusions

In this study, we analyzed transcriptome features based on the NGS and SMRT sequencing of the FER and STE ovules, in different periods in *P. tabuliformis*. Our findings showed that, compared with NGS, SMRT sequencing overcomes the limit of the length of reads, generating more transcriptome information. AS might play an important role in the growth and development of ovules, which may affect the fertility of *P. tabuliformis*, and provides valuable resources for SSR marker development. In addition, the low expression of DEGs related to auxin response in STE might lead to insufficient energy supply and nuclear division damage, which might be another reason for the scales being bigger in STE. Moreover, the DEGs associated with carbohydrate metabolism, signal transduction, mitosis, and apoptosis were determined, suggesting that the energy metabolism, regulation of cell cycle, and accumulation of reactive oxygen and lignin were abnormal in STE ovules. These findings may provide new insights into ovule abortion in plants, especially improving our understanding of the molecular mechanisms involved in the FNM process and ovule development in conifers (Figure 9).

## Figures and Tables

**Figure 1 ijms-22-03138-f001:**
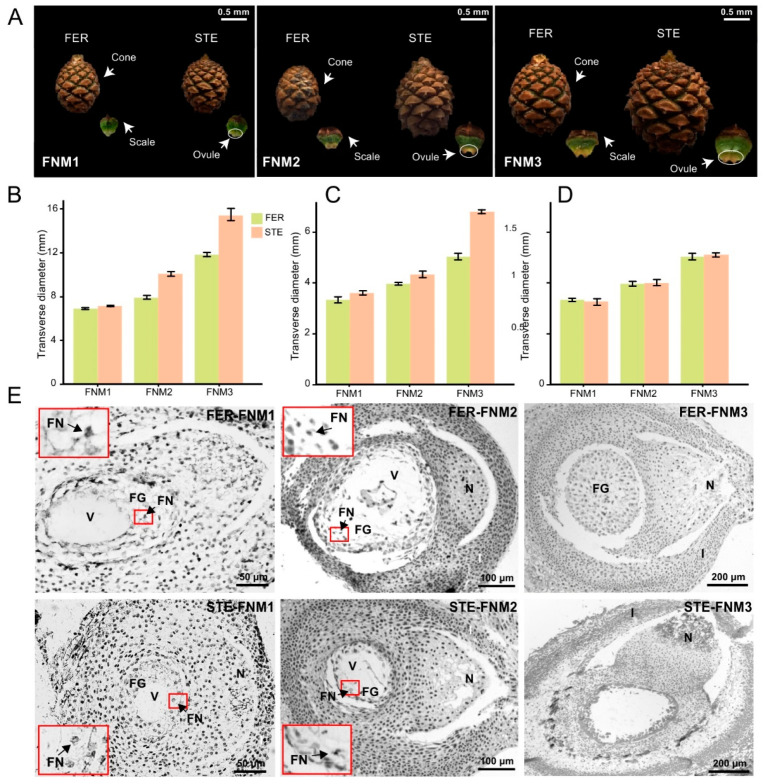
(**A**) Morphologic and microscopic observations of female fertile line (FER) and female sterile line (STE) ovules in *Pinus tabuliformis.* Cones and scales of different development stages. (**B**–**D**) The transverse diameters of cones, scales, and ovules in FER and STE. (**E**) The micrographs of ovules in different development. FG: female gametophyte; FN: free nucleus; I: integument; N: nucellus; V: vacuole.

**Figure 2 ijms-22-03138-f002:**
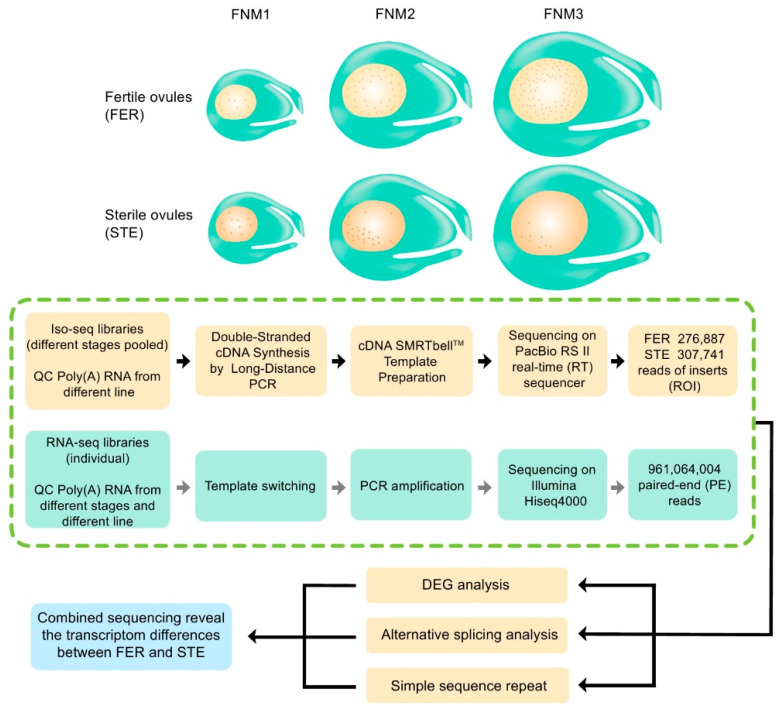
Flowchart of the experimental design and analysis for single-molecule real-time (SMRT) sequencing and RNA sequencing. Schematic graph of free nuclear mitosis (FNM) during ovule development in female fertile lines (FER) and sterile lines (STE) of *P. tabuliformis* collected for SMRT sequencing and Illumina sequencing. Combined PacBio and RNA-seq were used for a series of analyses, including alternative splicing (AS), differentially expressed gene (DEG) analysis, and simple sequence repeats (SSRs). This revealed the transcriptome differences between FER and STE.

**Figure 3 ijms-22-03138-f003:**
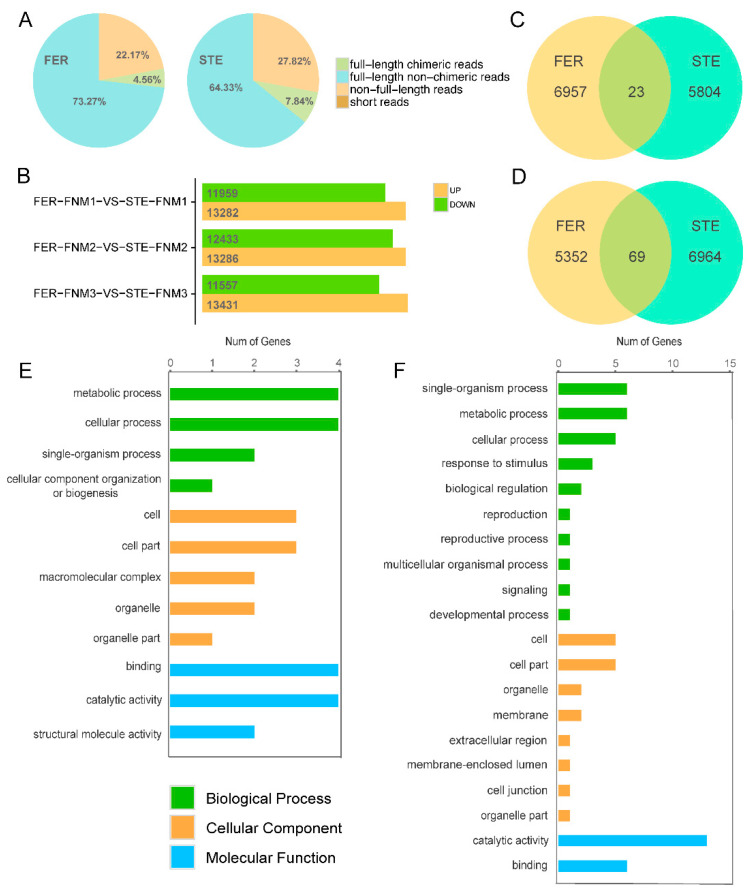
(**A**) Summary of SMRT sequencing proportions of four kinds of the circular consensus sequence (CCS), classified into full-length chimeras reads, full-length nonchimeric reads (FLNC), non-full-length reads, and short reads. (**B**) DEGs in each FNM period between FER and STE; (**C**) DEGs with opposite expression patterns during the FNM process in STE and FER, DEGs upregulated in FER and downregulated in STE; (**D**) DEGs upregulated in STE and downregulated in FER; (**E**) Gene Ontology (GO) classification of DEGs both upregulated in FER and downregulated in STE; (**F**) GO classification of DEGs both upregulated in STE and downregulated in FER.

**Figure 4 ijms-22-03138-f004:**
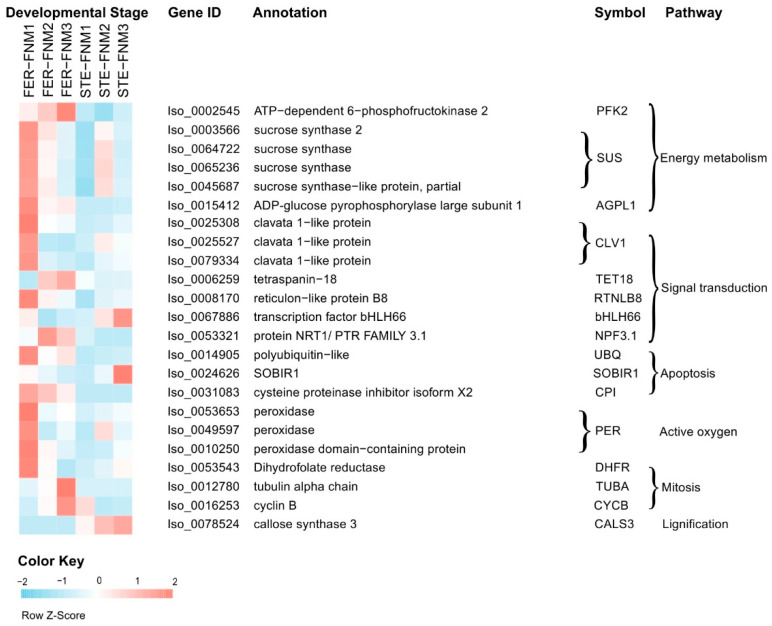
Heat map diagram of expression levels of DEGs, showing obvious differences in different lines. The gene expression level data were normalized to Z-score.

**Figure 5 ijms-22-03138-f005:**
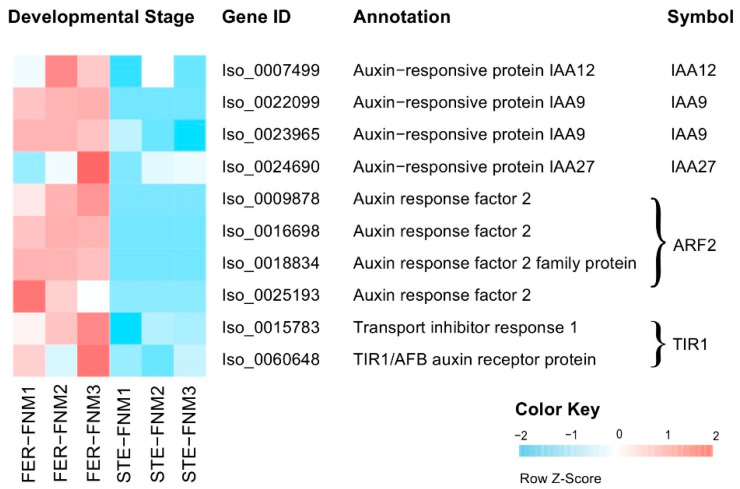
Heat map of DEGs expression levels associated with the auxin response. The gene expression level data were normalized to Z-score.

**Figure 6 ijms-22-03138-f006:**
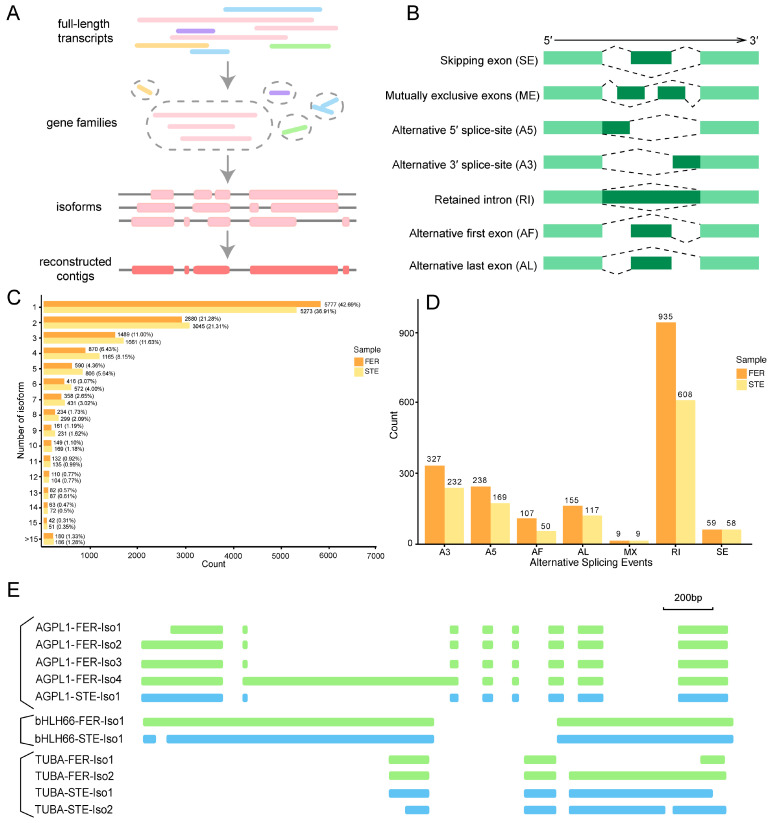
Comparison of different alternative splicing (AS) among FER and STE. (**A**) Schematic diagram of Cogent assembly; (**B**) visualization of seven AS modes; (**C**) distribution of isoform numbers in FER and STE; (**D**) different types of AS events in FER and STE; (**E**) the model graph shows AGPL1/bHLH66/TUBA generating different transcript isoforms of RI detected in two tissues. A3: alternative 3′ splice-site; A5: alternative 5′ splice-site; AF: alternative first exon; AL: alternative last exon; MX: mutually exclusive exons; RI: retained intron; SE: skipping exon.

**Figure 7 ijms-22-03138-f007:**
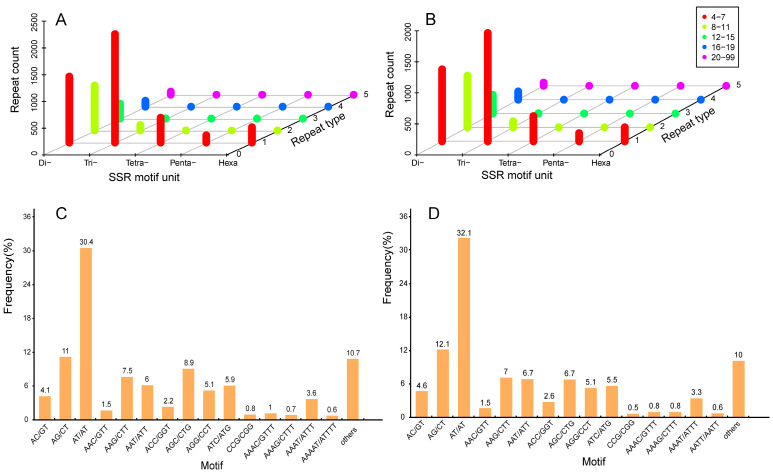
Comparison of different simple sequence repeat (SSR) among FER and STE. The distribution of SSRs based on the number of repeat units in FER (**A**) and STE (**B**); frequency distribution of SSRs based on motif types in FER (**C**) and STE (**D**).

**Figure 8 ijms-22-03138-f008:**
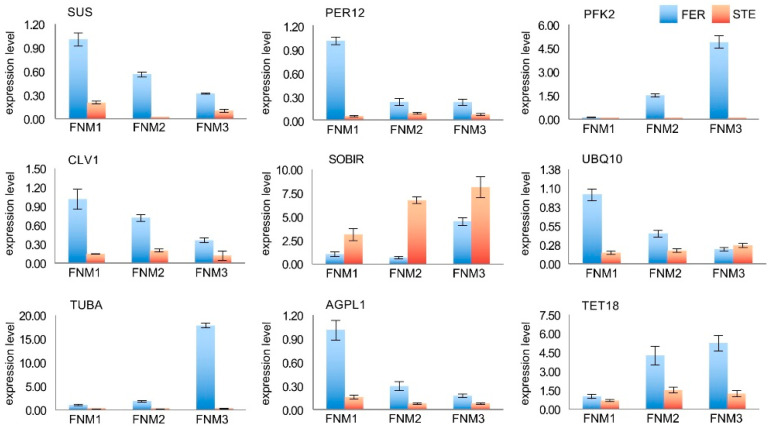
Candidate DEGs’ expression levels, as revealed by qRT-PCR. Sucrose synthase (SUS); peroxidase 12 (PER12); ATP-dependent 6-phosphofructokinase 2 (PFK2); clavata 1-like protein (CLV1); leucine-rich repeat receptor-like serine/threonine/tyrosine-protein kinase SOBIR1 (SOBIR1); polyubiquitin-like 10 (UBQ10); tubulin alpha chain (TUBA); ADP-glucose pyrophosphorylase large subunit 1 (AGPL1); tetraspanin-18 (TET18).

**Figure 9 ijms-22-03138-f009:**
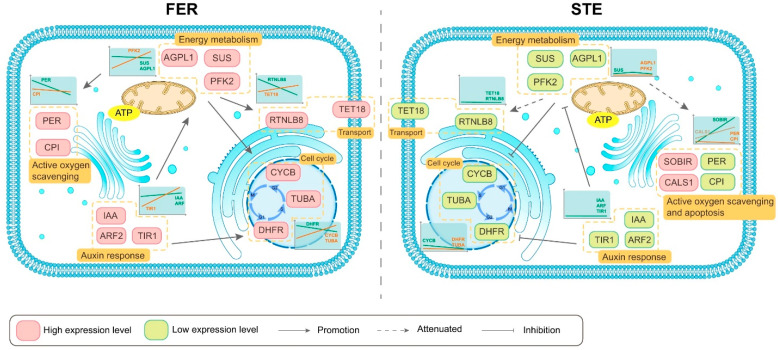
Speculative network model revealing the ovule abortion in a sterile female line of *Pinus tabuliformis* Carr. SUS: Sucrose synthase; AGPL1: ADP-glucose pyrophosphorylase large subunit 1; PFK2: ATP-dependent 6-phosphofructokinase 2; TET18: Tetraspanin-18; RTNLB8: Reticulon-like protein B8; CLV1: Clavata 1-like protein; CYCB: Cyclin B; TUBA: Tubulin alpha chain; DHFR: Dihydrofolate reductase; SOBIR1: Leucine-rich repeat receptor-like serine/threonine/tyrosine-protein kinase SOBIR1; CPI: Cysteine proteinase inhibitor; PER: Peroxidase; CALS3: Callose synthase 3; IAA: Auxin-responsive protein IAA; ARF2: Auxin response factor 2; TIR1: Transport inhibitor response 1.

**Table 1 ijms-22-03138-t001:** Summary of SSRs identified in *P. tabuliformis* transcriptome of FER and STE ovule.

Searching Item	Numbers	
	FER	STE
Total number of sequences examined	40,828	44,195
Total size of examined sequences (bp)	112,127,427	98,231,600
Total number of identified SSRs	5530	5096
Number of SSR-containing sequences	4276	3948
Number of sequences containing more than one SSR	868	796
Number of SSRs present in compound formation	475	487
Dinucleotide	2516	2485
Trinucleotide	2132	1846
Tetranucleotide	463	407
Pentanucleotide	134	130
Hexanucleotide	285	228

## Data Availability

All raw data of high-throughput sequencing have been deposited to the National Genomics Data Center (https://bigd.big.ac.cn) with the dataset accession number CRA004027.

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
