# Peer review of "Combined Transcriptome Analysis Reveals the Ovule Abortion Regulatory Mechanisms in the Female Sterile Line of Pinus tabuliformis Carr."

_ijms, 2021, doi:10.3390/ijms22063138_

Round 1

Reviewer 1 Report

This study addresses an important biological question, how the arrest or continuation of female gametophyte development during its mitotic phase is controlled, a process largely ignored in gymnosperms. The authors generated an impressive set of transcriptomic data, using state of the art RNAseq and long read sequencing (PacBio) methods, in woody plants very challenging to study. These resources are important as references for functional studies, especially in Pinus. And this study indeed identifies candidate genes that could be further analysed, for instance in terms of dynamics of expression patterns during ovule development in both normal and abortive ovules.

However I have found the manuscript very difficult to read, partly due to language and partly from a lack of clarity in how the combination of the two approaches is done to obtain a quantitative analysis; and overall, in my opinion, the analysis does support the conclusions. This might be due to either the pooling of the staged samples for the SMRT, or to the difference in genetic backgrounds in the fertile and sterile trees (despite a genetic study suggesting their relatedness). Indeed, less than 1% of the transcripts identified are common to the two compared samples (Fig3 D,E) and could be analysed as DEG. This low percentage is not discussed by the authors and questions the possibility to perform exhaustive transcriptomic comparisons using this material. And the main findings summarized in Fig9 rather suggest that apoptosis already started in the sterile ovules analysed, which is a consequence of abortion and does not shed light on the developmental mechanisms leading to this abortion. Focusing the analysis the first stages of gametogenesis (FNMM1 or earlier) should provide mechanistic information.

The aspect of alternative splicing is interesting, however, the work could describe the alternative splicing events found in only 3 DEGs between fertile and sterile ovule libraries. The biological significance of the importance AS on ovule sterility is unclear since different isoforms are found in both FER and STE samples for 2 genes, and only 1 gene displayed a single isoform in FER which is differently spliced in STE. These data do not support the conclusion in the discussion (line 401): « We calculated that the quantity difference of AS events in varying lines, especially the different isoforms of AGPL1, bHLH66, and TUBA between two lines might lead to the development of abnormal ovules in STE.” Further functional studies focusing on the different isoforms would be required.

In addition, the purpose of studying SSR is not clear here. Is it to develop transcriptomic markers (how would it be used to complement quantitative analysis by RNAseq data, which provides as well molecular markers?), or to generate genomic markers?

In terms of conclusions, I have also noted in the discussion that the authors suggest from their measurements that the growth of the cones and scales is more rapid in FER than STE, yet they finally conclude that « the viability of STE might be reduced and their growth and development blocked » which seems contradictory. Maybe rephrasing is needed.

In general, in both introduction and conclusion, the literature cited could be more focused on the precise developmental stages studied (ie gametogenesis), and some terms like “ovule development” are so general that the meaning remain unclear. As an example, the following discussion paragraph line 378 is very general: « Phenotypic changes were usually caused by changes at the molecular level, thus, accurate panorama of gene expression at the transcriptional level is particularly important. For example, the ovule development in rice was regulated by an MADS-box gene, the ovaries of osmads21 mutant were clearly swollen [35]. In Arabidopsis the RNAhelicase was required for ovule development, and knock down eIF4A-1 ovules would lose their nucellus, and these embryo sacs were empty [36]”. Functional genetic studies directly related to the specific gametogenesis mitotic stage exist, in Arabidopsis notably, and would be more appropriate to support the findings. Furthermore, some interpretations of the literature appear rather puzzling. For instance in the introduction lines 61-62: “however, silence TEs in the female gametophyte influenced the formation of megasporocyte [10,11].” is incorrect because megasporogenesis precedes the formation of the female gametophyte. In the discussion, « (…)CYCB high expressed during the cell cycle and degrade in mitosis, thus controlling enter and retreat from M phase in Nicotiana sylvestris [59].” The description is incorrect since CYCB1,1 gene rather peaks in M phase and is degraded upon M phase exit. Another example is the conclusions on TUBA, focusing on its potential role in mitosis (cytokinesis), when microtubules can act as well in the cell cortex, controlling cell expansion in the ovule.

Additional minor comments:

-Figure numbering is erroneous in the text (for instance Fig5 in text refers actually to Fig6;

Fig6 in text relates actually to Fig7, etc).

- In Fig 5D, the abbreviation used in x axis of the graph should be retaken/explained in Fig5B to help the reader.

- Line 286: in this paragraph, authors describe the alternative splicing events found in fertile and sterile ovule libraries. Fig5D show an apparently significant difference (yet no statistics are provided) in the frequency of Intron Retention (RI) between fertile/sterile; however this result is not commented in the text.

-In the Supplemental figures, the panels should be labelled to indicate the samples (or the legend adjusted to clarify).

Author Response

Thank you very much for your positive comments with valuable comments. In the past few days, we have seriously revised the manuscript. As you will see, the issues raised have been taken into account in the revised manuscript. Below we explain in details how they were addressed.

Question 1: Less than 1% of the transcripts identified are common to the two compared samples (Fig3 D,E) and could be analysed as DEG. This low percentage is not discussed by the authors and questions the possibility to perform exhaustive transcriptomic comparisons using this material.

Reply: We thank the reviewer for the expert comments. In this study, we got more than 20000 DEGs in each FNM (free nuclear mitosis) stage. To obtain the most significant DEGs, 23 DEGs were up-regulated in FER also down-regulated in STE (Figure 3B). 69 DEGs were up-regulated in STE as well as down-regulated in FER (Figure 3C). We found that 99 DEGs were up-regulated in one line and down-regulated in the other during all the three process of FNM, which might be the core factors of ovule abortion in P. tabuliformis. Therefore, the low percentage of DEGs were selected for further analysis. Based on your suggestions, we have added this part into Discussion in the revised manuscript.

Question 2: The main findings summarized in Fig9 rather suggest that apoptosis already started in the sterile ovules analysed, which is a consequence of abortion and does not shed light on the developmental mechanisms leading to this abortion. Focusing the analysis the first stages of gametogenesis (FNMM1 or earlier) should provide mechanistic information.

Reply: Thanks a lot for the reviewer’s comment. From the phenotypes and morphological analysis of ovules (Fig.1), we found that the anatomical structures of the STE and FER ovules were similar in the FNM1 stage, which became different from the FNM2 stage. Therefore we hypothesize that some key gene changes may start from the FNM2 period, but we can't exclude that some gene expression change from the first period or earlier. We fully agreed the comment of the review. According to our qPCR results in Figure 8, it has indicated that the expression level of SOBIR was higher in STE than in FER from FNM1 to FNM3. And the overexpression of SOBIR1 will lead to activation of cell death. These results showed that the high expression of SOBIR in the FNM1 stage might be a reason of apoptosis. Based on your comments, we have revised the details in Discussion in the revised manuscript.

Question 3: The biological significance of the importance AS on ovule sterility is unclear since different isoforms are found in both FER and STE samples for 2 genes, and only 1 gene displayed a single isoform in FER which is differently spliced in STE. These data do not support the conclusion in the discussion (line 401): « We calculated that the quantity difference of AS events in varying lines, especially the different isoforms of AGPL1, bHLH66, and TUBA between two lines might lead to the development of abnormal ovules in STE.” Further functional studies focusing on the different isoforms would be required.

Reply: Thanks for the reviewer’s comment. Yes, we agreed that the AS was an important event that affected all periods of plant development, which even affected the male sterile in rice (Sun et al.). Indeed, from our study, we found that the AS events were significantly different in two lines, and we further analyzed the AS events for DEGs with different expression patterns during the ovule development in the two lines (DEGs obtained from Fig 3B and 3C). From these DEGs, we got 3 genes with different isoforms in FER and STE. For these reasons, we speculated the conclusion that AS might play an important role in the growth and development of ovules which may be affected the fertility in P. tabuliformis. We just put forward a possibility, not a determined conclusion. As you suggested, the further functional studies are indispensable needed to get a definitive conclusion.

Sun, L.; Zhang, L.; Zhang, P.; Yang, Z.; Zhan, X.; Shen, X.; Zhang, Z.; Hu, X.; Xuan, D.; Wu, W.; Cao, L.; Cheng, S., Characterization and Gene Mapping of an Open Hul Male Sterile Mutant ohms1 Caused by Alternative Splicing in Rice. Chinese Journal of Rice Science 2015, 29, (5), 457-466.

Question 4: The purpose of studying SSR is not clear here. Is it to develop transcriptomic markers (how would it be used to complement quantitative analysis by RNAseq data, which provides as well molecular markers?), or to generate genomic markers?

Reply: Yes, in this manuscript the analysis of SSR is indeed to develop transcriptomic markers. We are sorry for this unclear expression. Although we can develop SSR markers rely on next-generation sequencing (NGS), but because NGS generally requires the assembly of short RNA-seq reads, the construction of transcriptomic sequences using NGS may be misassembled. Using SMRT sequencing, intact RNA molecules can be sequenced without the need for fragmentation or post-sequencing assembly, thus we can obtain the message of SSRs more accurately (Zeng et al, 2018). Accordingly, we have added these details and highlighted in the revised manuscript.

Zeng, D.; Chen, X.; Peng, J.; Yang, C.; Peng, M.; Zhu, W.; Xie, D.; He, P.; Wei, P.; Lin, Y.; Zhao, Y.; Chen, X., Single-molecule long-read sequencing facilitates shrimp transcriptome research. Scientific Reports 2018, (8), 16920.

Question 5: In terms of conclusions, I have also noted in the discussion that the authors suggest from their measurements that the growth of the cones and scales is more rapid in FER than STE, yet they finally conclude that « the viability of STE might be reduced and their growth and development blocked » which seems contradictory. Maybe rephrasing is needed.

Reply: Thanks for your comment. In this manuscript, we measured the size of the cones, scales and ovules for different free nuclear mitosis periods of FER and STE, and found that the size of the cone and scale was significantly larger in STE than in FER, however, the size of the ovules was similar in two lines (Fig 1A-1D). Additionally, the anatomical structures of the ovules showed that free nuclear mitosis was arrested in the FNM2 period. Thus we concluded that « the viability of STE might be reduced and their growth and development blocked ».

Question 6: In general, in both introduction and conclusion, the literature cited could be more focused on the precise developmental stages studied (ie gametogenesis), and some terms like “ovule development” are so general that the meaning remain unclear.

Reply: Many thanks for pointing out this deficiency. According to your suggestions, we have revised some vagueness expression in the section of Introduction and Discussion with the description more accuracy in the revised manuscript.  

“Huang indicated that the expression of cyclin genes affects the megaspore mother cell proliferation in Arabidopsis, which regulates the cell cycle progression and cell division.” “In Arabidopsis the ovules would lose their nucellus if eIF4A-1 (the RNA helicase) was knocked down, and these embryo sacs were empty.”

Minor:

1. Figure numbering is erroneous in the text (for instance Fig5 in text refers actually to Fig6; Fig6 in text relates actually to Fig7, etc).

Reply: We apologized for referring to the wrong figure number in original manuscript. “Fig. 5A-C” in Page. 9 was referred to “Fig. 6A-C”,  “Fig. 5D-E” and “Fig. 6A-D” in Page. 10 was referred to “Fig. 6D-E” and “Fig. 6A-D”. Accordingly, we have corrected these mistakes in the revised manuscript.

2. In Fig 5D, the abbreviation used in x axis of the graph should be retaken/explained in Fig5B to help the reader.

Reply: We apologized for this deficiency in the original manuscript. As you suggested, we have explained the abbreviation used in x axis of Figure 5D in Figure 5B (in the revised manuscript is 6D and 6B). Accordingly, the abbreviations were also defined in the legend of Figure 6 in the revised manuscript.

3. In this paragraph, authors describe the alternative splicing events found in fertile and sterile ovule libraries. Fig5D show an apparently significant difference (yet no statistics are provided) in the frequency of Intron Retention (RI) between fertile/sterile; however this result is not commented in the text.

Reply: Sorry for the incomplete analysis here. We have additional analyzed the details of the differences of AS modes in varying lines (Figure 6D) and completed the legend of Figure 6 (the figure showed the comparison of different AS among FER and STE). We have revised these details and highlighted in the revised manuscript.

“In each AS modes, there were more AS events in FER, and the retained intron (RI) was the most significant AS mode both in FER (51.09%) and STE (48.91%), which also the predominant mode in plants.”

4. In the Supplemental figures, the panels should be labelled to indicate the samples (or the legend adjusted to clarify).

Reply: Thanks for your comment. We have revised the legends of supplemental figures in the revised manuscript.

Reviewer 2 Report

In the work entitled “Single Molecular Real-Time Sequencing Revealing the Ovule Abortion Regulatory Mechanisms in the Female-Sterile Line of Pinus tabuliformis Carr.” the authors analyzed the transcriptome data of naturally occurring ovule abortion mutant line and the wild-type of Pinus tabuliformis Carr. With the aim of evaluating the mechanism of ovule abortion during the process of free nuclear mitosis. Single molecule real-time (SMRT) sequencing and next-generation sequencing (NGS) were the techniques used in this work.

The experiments are well designed, the manuscript is well organized, and the figures are of good quality. The all text requires a thorough review of the English. Every single sentence has grammatical errors what makes the reading quite difficult.

I will pinpoint some errors, but all the manuscript needs grammar review (for example, I will not go through the discussion section).

Review:

Line 29-30 – This sentence cannot be standing alone and should not start with “9”.

Line 42 – Replace “Cash crops” by “agronomically” important crops”

Line 48 – Replace “in embryos sacks lack” by “in lacking of embryo sacks development”…

Line 49 – Remove “of” after “abnormal

Line 52 – Replace “various” by “diverse”

Line 59 – “low expression” of what? Needs to be specified.

Line 64-66 – Review the sentence. It does not make sense.

Line 68 – Replace “limited” by “limits”

Line 70 – “Depending”, not “depend”

Line 70-76 – Please, rewrite the sentences. They are grammatically incorrect.

Line 76-78 – The authors ought to elaborate more on this idea; “AS is related to vegetative growth”. This is very vague and to some extent, not true since AS affects all levels of plant development and stress responses.

Line 79-80 – Please, do not introduce this short description of Ohms1 gene without any further introductory information. Also, the sentence is grammatically incorrect.

Line 84 – “gene caused female sterile”. This is not written properly.

Line 87-88 – Each characteristics? The authors must enumerate at least some of them.

Line 89 – “momentous”? Please, be more precise.

Line 97 – Review the grammar.

Line 98 – The authors wrote; “analyzed above”. What does this mean?

Line 100-101 and 102-105 – Review the grammar

Line 118-121 – Please remove. Doesn´t add anything to the work or the manuscript.

Line 130 – 132 – Review the English.

Line 132 – The authors start the sentence by saying “the difference” but then, does not describe which differences are present between the two lines.

Line 145-146 – The English is very confusing.

Line 149-152 – Please, don´t say “above periods”. Write something like: “the collection periods described above”…

Line 156 – I don’t understand the end and beginning of these two sentences. Review the English.

Line 162-164 – This sentence is very confusing.

Line 188-189 – The authors wrote; “FER also down-regulated”. I don’t get this sentence. “…also” (?) Please, describe the graph in a scientific fashion way.

Line 197-215 – Requires a deep review of the English. It’s very difficult to understand what the authors want to communicate.

Line 233-234 – “In both two parts”. This is a redundancy. The authors either write both or two.

Line 234-236 – It’s impossible to understand this sentence.

Line 239-241 – “combining two parts of DEGs”? what does this mean? Two parts of what and what is the criteria for such selection? The authors cannot write like this. To my knowledge, figures and tables do have not have the capacity to list anything. People do. This all sentence is a mess.

Fig. 4 and 5 legends – Review the English.

Line 516-517 and 523-524 – Review the grammar/English.

Line 525 – “Zhang et al performed paraffin sections” (?). Please write “paraffin sections were done according to Zhang et al.,”

Line 536 – “value higher”, not “value larger”

Line 536 – “pooled”, not “mixed”

Line 541 – “reads of the cDNA libraries”.

Line 550 – Please write “For annotation, sequences were BLAST”

Line 572 – Add full stop after “families”.

Line 573-574 and 576-580 – Review the grammar.

Line 582 – Did the authors used the same mRNA (cDNA) that was used to produce the libraries for sequencing? Specify this detail because it is important.

Author Response

Thank you very much for your positive comments with valuable comments. In the past few days, we have seriously revised the manuscript. As you will see, the issues raised have been taken into account in the revised manuscript. Below we explain in detail how they were addressed.

Reply to reviewer II

Point 1: Line 29-30 – This sentence cannot be standing alone and should not start with “9”.

Reply: We apologized for this unsuitable expression. And we have rewritten this sentence in the revised manuscript.

Point 2: Line 42 – Replace “Cash crops” by “agronomically” important crops”.

Line 48 – Replace “in embryos sacks lack” by “in lacking of embryo sacks development”…

Line 49 – Remove “of” after “abnormal.

Line 52 – Replace “various” by “diverse”.

Reply: Thanks a lot for pointing out these issues. And we have corrected these unsuitable expression in the revised manuscript.

Point 3: Line 59 – “low expression” of what? Needs to be specified.

Reply: Many thanks for pointing out this deficiency. The low expression here suggested the genes mentioned in the previous sentence, and we have completed this sentence in the revised manuscript.

Point 4: Line 64-66 – Review the sentence. It does not make sense.

Reply: Sorry for the grammar mistake here, we have rewritten this sentence in the revised manuscript.

Point 5: Line 68 – Replace “limited” by “limits”.

Line 70 – “Depending”, not “depend”.

Reply: We fully agreed with your comments, and we have corrected these words.

Point 6: Line 70-76 – Please, rewrite the sentences. They are grammatically incorrect.

Reply: We apologized for the unsuitable expression. And this sentence has been revised.

Point 7: Line 76-78 – The authors ought to elaborate more on this idea; “AS is related to vegetative growth”. This is very vague and to some extent, not true since AS affects all levels of plant development and stress responses.

Reply: We fully agreed with your comments. We have already mentioned AS can control the entire developmental pathway in the previous manuscript. As suggested, we have enriched the describe of the alternative splicing in the Introduction section in the revised manuscript.

Point 8: Line 79-80 – Please, do not introduce this short description of Ohms1 gene without any further introductory information. Also, the sentence is grammatically incorrect.

Reply: We are sorry for this unprofessional expression. We have revised this error and this sentence has been rewritten.

Point 9: Line 84 – “gene caused female sterile”. This is not written properly.

Reply: Sorry for the unsuitable expression. We have corrected the expression here in the revised manuscript.

Point 10: Line 87-88 – Each characteristics? The authors must enumerate at least some of them.

Reply: Many thanks for pointing out this deficiency. “Gymnosperms represent critical extant transitional groups in the change from the gametophyte to the sporophyte as the independent free-living generation, which has the most elaborate sporophyte and the most reduced gametophyte.” We have revised the section of Introduction in the revised manuscript to explain that gymnosperms was the node of plant evolution.

Point 11: Line 89 – “momentous”? Please, be more precise.

Reply: We are sorry for this unsuitable expression, this word has been reworded in the revised manuscript.

Point 12: Line 97 – Review the grammar.

Reply: We are sorry for the unsuitable expression. And this sentence has been revised.

Point 13: Line 98 – The authors wrote; “analyzed above”. What does this mean?

Reply: We are sorry for this unclear expression. We want to describe that our research team has analyzed this scientific problem based on the NGS platform, but used the wrong preposition. This sentence has been corrected in the revised manuscript.

Point 14: Line 100-101 and 102-105 – Review the grammar.

Reply: Sorry for this mistake and we have reworded the sentences in the revised manuscript.

Point 15: Line 118-121 – Please remove. Doesn´t add anything to the work or the manuscript.

Reply: We are sorry for this mistake and these sentences have been removed.

Point 16: Line 130 – 132 – Review the English.

Reply: We apologized for the unsuitable expression. And this sentence has been revised.

Point 17: Line 132 – The authors start the sentence by saying “the difference” but then, does not describe which differences are present between the two lines.

Reply: Thank you for pointing out our negligence, “the difference” here means that  the free nuclei were still increased in the FNM2 period of n FER which arrested in STE. We have corrected and highlighted this sentence in the revised manuscript.

Point 18: Line 145-146 – The English is very confusing.

Reply: Sorry for the grammar mistake here, we have rewritten this sentence in the revised manuscript.

Point 19: Line 149-152 – Please, don´t say “above periods”. Write something like: “the collection periods described above”…

Reply: We fully agreed with your comments, and we have corrected this phrase.

Point 20: Line 156 – I don’t understand the end and beginning of these two sentences. Review the English.

Line 162-164 – This sentence is very confusing.

Line 188-189 – The authors wrote; “FER also down-regulated”. I don’t get this sentence. “…also” (?) Please, describe the graph in a scientific fashion way.

Line 197-215 – Requires a deep review of the English. It’s very difficult to understand what the authors want to communicate.

Reply: We are sorry for the unprofessional expression, and we have reworded these sentences in the revised manuscript.

Point 21: Line 233-234 – “In both two parts”. This is a redundancy. The authors either write both or two.

Reply: Yes, we agreed with your expert comments. We have deleted “both” as suggested.

Point 22: Line 234-236 – It’s impossible to understand this sentence.

Line 239-241 – “combining two parts of DEGs”? what does this mean? Two parts of what and what is the criteria for such selection? The authors cannot write like this. To my knowledge, figures and tables do have not have the capacity to list anything. People do. This all sentence is a mess.

Fig. 4 and 5 legends – Review the English.

Line 516-517 and 523-524 – Review the grammar/English.

Reply: We are sorry for the unprofessional expression. We have revised this error and this sentences have been rewritten.

Point 23: Line 525 – “Zhang et al performed paraffin sections” (?). Please write “paraffin sections were done according to Zhang et al.,”

Line 536 – “value higher”, not “value larger”.

Line 536 – “pooled”, not “mixed”.

Line 541 – “reads of the cDNA libraries”.

Line 550 – Please write “For annotation, sequences were BLAST”.

Line 572 – Add full stop after “families”.

Reply: Yes, we agreed with your expert comments. According to your seggestion, we have reworded the sentence in the revised manuscript as suggested.

Point 24: Line 573-574 and 576-580 – Review the grammar.

Reply: We apologized for the unsuitable expression. And these sentences have been revised.

Point 25: Line 582 – Did the authors used the same mRNA (cDNA) that was used to produce the libraries for sequencing? Specify this detail because it is important.

Reply: Many thanks for pointing out this deficiency. Yes, the mRNA samples used for real-time quantitative PCR analysis were same with the mRNA samples for sequencing. And we have add this detail and highlighted in the revised manuscript.

Reviewer 3 Report

Gong et al provide a real-time insight that appeared to be missing in the field on how splicing governs ovule development and fertility also in gymnosperms. The article entitled Single Molecular Real-Time Sequencing Revealing the Ovule Abortion Regulatory Mechanisms in the Female-Sterile Line of Pinus tabuliformis Carr. is scientifically interesting and attempted to describe the mechanism through which ovule abortion occurs in gymnosperms, however, fell slightly short of its title claim. With a good effort on a revised manuscript, this work will be informative to the community.

Comments that should be addressed (major);

  1. Authors found five cellular processes to be differentially regulated between STE and FER lines. However, the authors elaborated simply on auxin without providing/discussing a linkage between auxin and ovule abortion.
  2. how are the findings of this manuscript compared to the findings in other proposed ovule abortion models?
  3. Figure 1E, it is quite difficult to make out the differences between the pointed structures, FG vs FN. Authors should make an effort to improve the figure outline, for example, they could use an illustrative cartoon of the same/representative images to help highlight the difference.
  4. For Figure 6, authors should provide a semi RT-PCR gels of representative candidate genes showing the several forms of missplicing presented in Fig. 6B. This will provide a semi quantitative insight on the dominancy of the different missplicing events

Minor comments

1.The introduction is quite fragmented needs better transition between paragraphs. More importantly, the introduction is missing background on splicing and its contributing role in development to be appreciated.

-line 118-121, the authors forgot to delete journal template instructions?

2. It will help significantly to have the English revised. Some sentences are very difficult to understand.

3. It is not necessary to intriduce new abbreviations in Fig 1 and Fig 2(FL and SL) which are also not defined.

4. Lines 161-164, poorly written, not clear at all.

5. Whole Fig. 3 should be moved to supplementary except D/E/G/H

6. Some of the text on Fig. 3 are a bit too small. Fig 3G/H should be plotted together (clustered columns) for direct comparison.

7. Lines 261-262, NGS cant be used to detect auxin, please correct

8. Fig. 4 and Fig. 5, it is not immediately clear to me what the values represent on the heat-map.

9. Lines 279-281, authors should state how transcripts were grouped into gene families

10. Fig 4D, abbreviations are not defined

11. Figure 6, lines 282-283, is 4.44% vs 4.49% >10 isoforms more significant, why not other categories that appear to have even bigger differences? - authors should not be biased because of the large number of isoforms, in fact the larger number of isoforms are unlikely to be a relevant direct miss-splicing influence, whereas less frequent isoforms are more of a direct consequence.

12. Lines 305-312, I think it's a meaningless paragraph with no link or clearly defined objective. Authors should either introduce and clearly link microsatellites and ovule abortion or at least introduce their hypothesis on microsatellite roles on ovule development.

13. Line 344, should be Fig 8 not Fig 7

14. Fig 8 is missing y-axis title, how was the qPCR data normalized?

15. Lines 467-480, i completely struggled to understand here. 

16. I would suggest to move the highly speculative model of Fig 9 to supplementary.

17. Fig S1, it will be useful to circle and label each triplicate within the PCA rather than using a long legend with overlaping coloration

18. Fig S3, is there an order how the profiles are organized/presented? or is it randomized?

Author Response

Thank you very much for your positive comments with valuable comments. In the past few days, we have seriously revised the manuscript. As you will see, the issues raised have been taken into account in the revised manuscript. Below we explain in details how they were addressed.

Question 1: Authors found five cellular processes to be differentially regulated between STE and FER lines. However, the authors elaborated simply on auxin without providing/discussing a linkage between auxin and ovule abortion.

Reply: Thank you for your expert comments. Accordingly to your suggestion, we have completed the discussion about the linkage between auxin and ovule development at the DISCUSSION section. “Efflux and influx carriers regulated the through of auxin and the distribution of auxin influenced the development of ovule (Fu et al., 2014). Robert et al. also found that local auxin source coupled to feedback regulation of auxin transporter facilitators polarity in the embryo was sufficient to generate a robust auxin gradient that instructs the formation of embryo in Arabidopsis” (Robert et al. 2020). And the references mentioned had been added in the part of REFERENCES in the revised manuscript.

Fu, W.; Zhao, Z.; Ge, X.; Ding, L.; Li, Z., Anatomy and transcript profiling of gynoecium development in female sterile Brassica napus mediated by one alien chromosome from Orychophragmus violaceus. BMC Genomics 2014, 19, (61), 2-15.

Robert, H; Park, C.; Gutierrez, C.; Wojcikowska, B.; Pencik, A.; Novak, O.; Chen, J.; Grunewald, W.; Dresselhaus, T.; Friml, J.; Laux, T, Maternal auxin supply contributes to early embryo patterning in Arabidopsis. Nature Plants 2018, 4(8):548-553.

Question 2: how are the findings of this manuscript compared to the findings in other proposed ovule abortion models?

Reply: Thanks for your expert comments. According to your suggestion, we compared the ovule abortion model with other spieces, and obtained the feature of the P. tabuliformis female sterile. We have added the discussion of the compare of other ovule abortion models and highlighted in the revised manuscript.

“Yang et al. reported that the ovule abortion mechanism of rice was related to starch and sucrose metabolism, plant hormone signal transduction, protein modification and degradation, oxidative phosphorylation (Yang et al.). In Brassica napus, DEGs regulating brassinosteroid biosynthesis, adaxial/abaxial axis specification, auxin transport and signaling were regarded as the reasons of female sterile (Fu et al). Similarly to other spieces, in our study the energy metabolism, auxin transport and signaling were also essential in the ovule abortion mechanism of P. tabuliformis. Additionally, during the FNM process, cell division and apoptosis might also be important elements. The reason for this difference perhaps was the free nuclear mitosis was the indispensable process of the ovule formation in gymnosperm.”

Yang, L.; Wu, Y.; Yu, M.; Mao, B.; Zhao, B.; Wang, J., Genome-wide transcriptome analysis of female-sterile rice ovule shed light on its abortive mechanism. Planta 2016, 244, (5), 1011-1028.

Fu, W.; Zhao, Z.; Ge, X.; Ding, L.; Li, Z., Anatomy and transcript profiling of gynoecium development in female sterile Brassica napus mediated by one alien chromosome from Orychophragmus violaceus. BMC Genomics 2014, 19, (61), 2-15.

Question 3: Figure 1E, it is quite difficult to make out the differences between the pointed structures, FG vs FN. Authors should make an effort to improve the figure outline, for example, they could use an illustrative cartoon of the same/representative images to help highlight the difference.

Reply: Many thanks for pointing out this deficiency. We have revised Figure 1E and enlarged the image of FN in the revised manuscript.

Question 4: For Figure 6, authors should provide a semi RT-PCR gels of representative candidate genes showing the several forms of missplicing presented in Fig. 6B. This will provide a semi quantitative insight on the dominancy of the different missplicing events

Reply: We thank the reviewer for the expert comments. We also think the results will be more convincing with a semi RT-PCR gels of representative candidate genes. Unfortunately, due to the COVID-19, the permission of our school was delayed and we are unable to enter the laboratory for experiments. To respect your suggestion,we have added related outlook in the section of DISSCUSION. This part will be an important research content for further studies in the revised manuscript.

Minor:

  1. The introduction is quite fragmented needs better transition between paragraphs. More importantly, the introduction is missing background on splicing and its contributing role in development to be appreciated.

-line 118-121, the authors forgot to delete journal template instructions?

Reply: We are grateful to the referee for pointing out this issue in our previous manuscript. We have reworded the transition between paragraphs an enriched the part of background of alternative splicing and its relation to the female sterile in the section of introduction.

And we are sorry for forgot to delete line 118-121 and these sentences have been removed.

  1. It will help significantly to have the English revised. Some sentences are very difficult to understand.

Reply: We are sorry for the grammar mistake in this manuscript. We have revised the sentences with unsuitable expression in the revised manuscript.

  1. It is not necessary to intriduce new abbreviations in Fig 1 and Fig 2(FL and SL) which are also not defined.

Reply: We apologized for referring to the wrong abbreviations in original manuscript.  We have corrected these mistakes in the revised manuscript.

  1. Lines 161-164, poorly written, not clear at all.

Reply: We are sorry for the unprofessional expression, and we have reworded these sentences in the revised manuscript.

  1. Whole Fig. 3 should be moved to supplementary except D/E/G/H

Reply: Thanks a lot for the your comment. According to your suggestion, we moved Figure 3B/C/F to the supplementary in the revised manuscript.

  1. Some of the text on Fig. 3 are a bit too small. Fig 3G/H should be plotted together (clustered columns) for direct comparison.

Reply: Many thanks for pointing out this deficiency. We have revised Figure 3 in the revised manuscript.

  1. Lines 261-262, NGS cant be used to detect auxin, please correct

Reply: Thank you for pointing out this issue, we have corrected this mistake in the revised manuscript.

  1. 4 and Fig. 5, it is not immediately clear to me what the values represent on the heat-map.

Reply: We are sorry for this deficiency. In this study, the gene expression level data were normalized to Z-score. After normalization we can limit the data in a smaller range, and compare the expression level of the same gene in different samples more intuitive. And we have added this detail in the figure legend in the revised manuscript.

  1. Lines 279-281, authors should state how transcripts were grouped into gene families

Reply: Thank you for pointing out our negligence, the full-length transcripts were partitioned into gene families based their k-mer similarity, and we have added this details in the section of Results and highlighted in the revised manuscript.

  1. Fig 4D, abbreviations are not defined

Reply: Thanks a lot for pointing out this issue. We think your mean is Figure 5D of the previous manuscript, and we have explained the abbreviation used in x axis of Figure 5D in Figure 5B (in the revised manuscript is 6D and 6B). Accordingly, the abbreviations were also defined in the legend of Figure 6 in the revised manuscript.

  1. Figure 6, lines 282-283, is 4.44% vs 4.49% >10 isoforms more significant, why not other categories that appear to have even bigger differences? - authors should not be biased because of the large number of isoforms, in fact the larger number of isoforms are unlikely to be a relevant direct miss-splicing influence, whereas less frequent isoforms are more of a direct consequence.

Reply: We are very sorry to the negligence here, we have counted the contigs with more than 15 isoforms and revised Figure 6C in the revised manuscript. Accordingly, we have revised the results related to Figure 6C.

  1. Lines 305-312, I think it's a meaningless paragraph with no link or clearly defined objective. Authors should either introduce and clearly link microsatellites and ovule abortion or at least introduce their hypothesis on microsatellite roles on ovule development.

Reply: Thank you for your comments. According to your suggestion, We have streamlined this paragraph and integrate it into the next paragraph.

  1. Line 344, should be Fig 8 not Fig 7

Reply: We apologized for referring to the wrong figure number in original manuscript. “Fig. 7” in Page. 12 was referred to “Fig. 8”. Accordingly, we have corrected these mistakes in the revised manuscript.

  1. Fig 8 is missing y-axis title, how was the qPCR data normalized?

Reply: Many thanks for pointing out these mistakes, Figure 8 has been revised in the revised manuscript, and y-axis of Figure 8 is “expression level”. Accordingly, EF1 (elongation factor 1) was selected as the internal control for normalizing the results, and the FER ovules in FNM1 stage were used as a reference sample whose value was set to 1. We have added the details of qRT-PCR in the section of Materials and Methods.

  1. Lines 467-480, i completely struggled to understand here.

Reply: We are sorry for this unprofessional expression. We have corrected the expression here in the revised manuscript.

  1. I would suggest to move the highly speculative model of Fig 9 to supplementary.

Reply: Thanks a lot for the your comment. Figure 9 was the summary of discussion, thus we think put Figure 9 in the main body of the article is more intuitive.

  1. Fig S1, it will be useful to circle and label each triplicate within the PCA rather than using a long legend with overlaping coloration

Reply: Yes, we agreed with your comments. According to your suggestion, we modified Figure S1 in the revised manuscript.

  1. Fig S3, is there an order how the profiles are organized/presented? or is it randomized?

Reply: Thanks a lot for the your comment. The profiles in Figure S3 was listed rely on p-value, from low to high. And the p-value was marked on each profile, we also added the description of Fig S3 in the revised manuscript. And the number of profiles was the representation of expression patterns.

Round 2

Reviewer 1 Report

Dear authors,

Thank you very much for the clarifications provided in your reply. You will find my comments in the file attached, in blue below each questions of your reply, and additional comments. I also recommand extensive english revision as the manuscript is very difficult to read and understand. 

Author Response

We have received your comments, and we are grateful for the comments and valuable suggestions. After reading the comments, we came to realize that we indeed had some weak points and unsuitable expression in our manuscript. We have revised the manuscript according to the comments from the reviewer point by point, and detailed corrections are marked in red highlighting in the revised manuscript. Additionally, this manuscript will be language-edited by MDPI before published.

Reply to authors for Question 1: Many thanks to the authors for the clarification. However, it is not clear from which analysis comes the “more than 20000 DEGs in each FNM stage” identified and now mentioned in the discussion. For the reader to understand properly, it should be explained how the information from the RNAseq and Long Read Sequencing pooled libraries were combined to obtain all the DEGs identified. Furthermore, the comparisons of all FER and STE libraries should be presented in a figure or sup figure, showing the total number of individual genes identified, the number of identified genes shared between FER and STE at each FNM stage (or in pooled stages for Long Read Sequencing), and finally the DEGs obtained from the combined analysis. This should be also described in the text in the result section.

Reply: Thank you for your expert comments. According to your suggestion:

i. We have described how the RNA-seq and Long-read sequencing were combined in the revised manuscript, and we have revised the section of Results. “Through SMRT sequencing, 19.07 GB of raw data were produced from two libraries (FER, STE), and 81,524 unique full-length transcripts were obtained through pipeline analysis. Moreover, 0.96 billion pairs of clean reads were produced from 18 libraries using NGS to calculate the expression level (reads per kilobase of exon model per million mapped reads, RPKM) of each full-length transcripts, with a mapping ratio of 85.43–87.99%.”

ii. In addition, we have added a figure (Figure 3B) to show the number of DEGs in FER and STE at different FNM (free nuclear mitosis) stages in the revised manuscript.

iii. We have completed the analysis about the DEGs between FER and STE, and counted the DEGs up- or down-regulated in each FNM stage. “To find out the DEGs which might influenced the development of FG, we counted the DEGs of each FNM period and these expression levels in FER and STE. From figure 3B, we found in all stages there were more DEGs up-regulated in FER than down-regulated. And there were about 25,000 DEGs in each FNM period. To obtain the most significant DEGs, according to the variety of gene expression during the whole development process in FER and STE, genes could be clustered into eight profiles by Short Time-series Expression Miner (STEM) software.”

Reply to authors for Question 2: Ovule abortion implies apoptosis. The overexpression of SOBIR1 from stage FNM1 likely represents an early marker of apoptosis, but it does not explain what is the regulatory mechanism triggering apoptosis (and therefore triggering ovule abortion). This triggering mechanism likely starts before the stage FNM1, as suggested by this study. Moreover, the observed downregulation of many genes in FER ovules might be a consequence of apoptosis, rather than inducing mechanisms. The study cannot distinguish between controlling (cause) and markers (consequence) genes, without further functional studies. This important distinction must be clearly explained in a paragraph in the discussion, and all claims on identification of regulatory mechanisms should be modulated as hypothetical (including in abstract). The title should be rephrased to present the work as a transcriptomic resource for ovule sterility in Pinus.

Reply: We agreed with your comments that the low expression level of SOBIR1 might be the the reason of ovule abortion, which also might cause female sterile in P. tabulaeformis. “However, the high expression level of SOBIR1 in STE also might be the result of ovule abortion, apoptosis occured in the FNM1 period led to the vicious circle that the development of FG damaged. And we found that many genes regulated the essential pathways of FG developing were down-regulated in STE, which also maybe induced by apoptosis. (in the section of Discussion)” In addition, to make our results more credible, we modulated the sentences about hypothetical regulation mechanisms in the revised manuscript. “And the apoptosis happened in STE might be another reason which affected the expression level of these genes. (in the section of Abstract)” According to your suggestion, we have adjusted the title of this manuscript: “Combined Transcriptom Analysis Reveals the Ovule Abortion Regulatory Mechanisms in the Female-Sterile Line of Pinus tabuliformis Carr.”

Reply to authors for Question 3: Thank you for your reply. To clearly state that AS is put forward as a possibility, not a functional conclusion, the conclusive sentence “We calculated that the quantity difference of AS events in varying lines, especially the different isoforms of AGPL1, bHLH66, and TUBA between two lines might lead to the development of abnormal ovules in STE.” should be reformulated, for instance as: “We speculate that quantitative differences in AS events, especially the different isoforms of AGPL1, bHLH66, and TUBA, between the two FER and STE lines might lead to the development of abnormal ovules in STE.”

Reply: We apologized for this unsuitable expression. We have reworded this sentence as you suggested: “We speculate that quantitative differences in AS events, especially the different isoforms of AGPL1, bHLH66, and TUBA between two lines might lead to the development of abnormal ovules in STE.”

Reply to authors for Question 4: I didn’t mean that SSRs could be developed from RNAseq data, but that RNAseq data can be used to provide transcriptomic markers (ie RT-PCR tests) as shown in your manuscript. Therefore, the purpose of this section should be clarified, notably by modifying the first sentence of the section: “To further investigate the differences of regulatory mechanisms between FER and STE ovules, thus the microsatellites were identified”. This link between regulatory mechanisms and microsatellites is unclear. In other parts of the text (discussion), some formulations in the text still suggest that the purpose is rather to develop genomic tools: for instance (line 422) “For example, SSRs marker has been used to identify the position of the gene caused female sterile in wheat and soybean [24,25].”. In conclusion, the general purpose of developing SSR markers must be clarified in the result section. Additionally, how this resource will be made available to the community?

Reply: We apologized for the unsuitable expression. We have revised the first sentence of SSR section: “To investigate the differences of regulatory mechanisms between FER and STE ovules, we discovered the characteristics of SSR markers in two lines.” And the misleading describes were removed in the revised manuscript. Additionally, all raw data of high-throughput sequencing have been deposited to the National Genomics Data Center (https://bigd.big.ac.cn) and would be released immediately after this paper published.

Reply to authors for Question 5: I do apologize for a mistake in writing my question, which probably made it unclear: it should be read “the authors suggest from their measurements that the growth of the cones and scales is more rapid in STE than FER”, as written in the current manuscript: Line 384: “However, till the FNM2 stage, STE were slightly larger than those of the FER, but larger than those of the FER significantly at FNM3 period, indicating that the growth of cones and scales was more rapidly in the STE than in the FER (Fig. 1).”. My question still stands: this finding suggests that growth and development of the cones and scales are not reduced nor blocked in FER. Moreover, the ovules are of similar size in the two lines, suggesting again that ovule growth is not blocked. Thus, the defects of the free nuclear mitoses do not seem to arise as a consequence of altered ovule growth, nor have consequences on it. Therefore, the conclusion should be modified to explain these observations.

Reply: Thanks for your expert comments. Yes, the size of ovules between FER and STE was similar in each FNM stage, the difference of them was the development of female gametophyte, not the ovule. Sorry for this unclear expression, and these misleading expressions were modified in the sections of Results and Discussion in the revised manuscript.

Reply to authors for Question 6: The proposed revision ““Huang indicated that the expression of cyclin genes affects the megaspore mother cell proliferation in Arabidopsis, which regulates the cell cycle progression and cell division.” still seems problematic for its unclear formulation and overall because it relates to megasporogenesis and not to megagametogenesis (see below). I still think introduction and conclusion can be improved by removing the references not focused which don’t bring functional information on the process studied here, and are confusing. For instance, in the introduction, the paragraph starting with: “Previous studies revealed that many genes might be associated with female sterility”, the examples listed after are not directly related to female sterility, and both the works on ago9 and by Huang et al relates to sporogenesis and not gametogenesis. etc. Indeed many genes have been identified that control ovule sterility, and to describe the state of the art, an alternative would be to cite reviews about the regulatory molecular mechanisms of ovule and gametogenesis in plants? Similarly, in the discussion, line 391, I would remove there the two examples of osmad1 (ref 38) and eIF4A-1 (ref 39), actually they are rather confusing as they don’t bring focused information in this paragraph explaining the general approach.

Reply: We are grateful to the referee for pointing out this issue in our previous manuscript. According to your suggestion, we have removed the references not focused on our background. Furthermore, we have cited reviews related to the development of ovules and female gametophyte and reworded this paragraph: “In the past few years, many studies about the molecular mechanism of ovule development have been reported, revealing many factors were related to the female sterile in plants. For example, Pagnussat et al found that the distribution of auxin during the female gametophyte development regulated the differentiation and conversion of gametic and the other cell, affected the embryo sac development in the initial stage[1]. Reactive oxygen also played a role in embryo sac patterning and fertilization, which influenced the development of female gametophyte by its concentration[2]. Moreover, ribosomal protein mutations might led to ribosome insufficiency, and the expression level of ribosomal protein genes was related to the number of seed and the defective ovules[3]. Studies about plant female sterile were mainly concentrated on herbs with short life cycle, but less on woody plants, especially gymnosperms.”

[1] Pagnussat, G.; Alandete-Saez, M.; Bowman, J.; Sundaresan, V., Auxin-Dependent Patterning and Gamete Specification in the Arabidopsis Female Gametophyte. Science 2009, 324, (5935), 1684-1689.

[2] Martín, M.; Distefano, A.; Zabaleta, E.; Pagnussat, G., New insights into the functional roles of reactive oxygen species during embryo sac development and fertilization in Arabidopsis thaliana. Plant Signaling & Behavior 2014, 8, (10), e25714.

[3] Zsogon, A.; Szakonyi, D.; Shi, X.; Byrne, M., Ribosomal Protein RPL27a Promotes Female Gametophyte Development in a Dose-Dependent Manner. Plant Physiology 2014, 165, (3), 1133-1143.

Minor:

i. - I have noticed that Data availability statements is missing, please indicate how the transcriptomic sequences generated (and the SSR markers potential resource) will be made available to the community.

Reply: Thank you for your suggestion. Sorry for this omission and and all raw data of high-throughput sequencing have been deposited to the National Genomics Data Center (https://bigd.big.ac.cn) and would be released immediately after this paper published.

ii. - Section 2-3: Please explain on which total set of genes was performed the functional annotation of full-length transcriptomes of FER and STE ovules.

Reply: Thanks for your comments. We combined with equal amounts of total RNA from three stages (FNM1, FNM2, and FNM3) in FER and STE. And we got two pooled libraries (FER and STE) subjected to an SMRT sequencing. The full-length transcripts obtained by SMRT were performed the functional annotation in section 2-3, and this point has been rewritten in the revised manuscript.

iii. - Section 2-4: Please indicate at the beginning on which set of genes the STEM analysis was performed.

Reply: We thank the reviewer for the expert comments. Due to SMRT sequencing would not analyze the expression level of genes, which mainly used for qualitative analysis. Thus the genes generated from NGS and corrected by SMRT were used for the STEM analysis. And we have revised the sentences about this point.

iv. - Figure S3: Please indicate how many genes were analyzed to obtain each profile, for each line.

Reply: Many thanks for pointing out this deficiency. We have added the number of genes of Figure S3 in the revised manuscript.

v. - Please explain in Methods how Z-score normalization was performed.

Reply: Thank you for pointing out this issue, z-score normalization was the expression level of each genes minus the average expression level of all genes, then divided by standard deviation. And we have enriched the methods of z-score normalization in the revised manuscript.

vi. - Line 265: Please correct sentence: “Auxin have been implicated in the nuclear division and cell growth. Based on the sequencing data, we determined the expression levels of auxin in FER and STE of different development stages. »: the analysis does not measure auxin level, rather the expression levels of genes implicated in the auxin pathway.

Reply: Sorry for the grammar mistake here, we have rewritten this sentence in the revised manuscript: “Based on the sequencing data, we found that the expression levels of genes related to auxin were diversity in FER and STE of different development stages.”

vii. -Please correct in the discussion, « (...)CYCB high expressed during the cell cycle and degrade in mitosis, thus controlling enter and retreat from M phase in Nicotiana sylvestris [59].” The description is incorrect since CYCB1,1 gene rather peaks in M phase and is degraded upon M phase exit.

Reply: We apologized for this unsuitable expression, this sentence was reworded in the revised manuscript.

Reviewer 3 Report

I accept the revised version, the authors made some needed corrections but unfortunately could not perform a supporting experiment for verification of their results. Despite, I believe it's of sufficient quality for publication at IJMS.

Said 

Author Response

We have received your comments, and we are grateful for your suggestions. Accordingly, we have further revised some details in this manuscript and marked in red highlighting in the revised manuscript. Additionally, this manuscript will be language-edited by MDPI before published.

Round 3

Reviewer 1 Report

The authors improved the manuscript taking into account my comments, and the article seems now suitable for publication in IJMS, providing revision of english which is still difficult to understand in this revised version.

Author Response

We have received your comments, and we are grateful for your suggestions.  And this manuscript have been language-edited by MDPI, the details revisions of language-editing were highlighted in the revised manuscript. Appendix was the certificate of language-editing.

Thanks again for your valuable comments for these rounds.

(Please see in attachment)
